# Molecular imaging of liver inflammation using an anti-VCAM-1 nanobody

Maxime Nachit[1,2,7], Christopher Montemagno[3,7], Romain Clerc[3], Mitra Ahmadi[3], François Briand[4], Sandrine Bacot[3], Nick Devoogdt[5], Cindy Serdjebi [6], Catherine Ghezzi[3], Thierry Sulpice[4], Alexis Broisat[3] ✉, Isabelle A. Leclercq [1,8] & Pascale Perret[3,8]

To date, a biopsy is mandatory to evaluate parenchymal inflammation in the liver. Here, we evaluated whether molecular imaging of vascular cell adhesion molecule-1 (VCAM-1) could be used as an alternative non-invasive tool to detect liver inflammation in the setting of chronic liver disease. To do so, we radiolabeled anti-VCAM-1 nanobody ($^{99m}$Tc-cAbVCAM1-5) and used single-photon emission computed tomography (SPECT) to quantify liver uptake in preclinical models of non-alcoholic fatty liver disease (NAFLD) with various degree of liver inflammation: wild-type mice fed a normal or high-fat diet (HFD), FOZ fed a HFD and C57BL6/J fed a choline-deficient or -supplemented HFD. $^{99m}$Tc-cAbVCAM1-5 uptake strongly correlates with liver histological inflammatory score and with molecular inflammatory markers. The diagnostic power to detect any degree of liver inflammation is excellent (AUROC 0.85−0.99). These data build the rationale to investigate $^{99m}$Tc-cAbVCAM1-5 imaging to detect liver inflammation in patients with NAFLD, a largely unmet medical need.

An estimated 1.5 billon persons have chronic liver disease (CLD) worldwide, most commonly due to non-alcoholic fatty liver disease (NAFLD; ~60%) or viral hepatitis (~38%)[1]. CLD is typically asymptomatic, at least until end-stages. Hence patients silently progress and eventually develop cirrhosis and/or hepatocellular carcinoma (HCC)[2]. Techniques to identify in situ chronic inflammation−the driving force of liver disease progression[3]−are lacking. At the present time, the only mean to diagnose liver inflammation is a biopsy. Beside the invasive nature of the technique and its potential complications (e.g., bleedings)[4], biopsies suffer from an inevitable sampling bias as they merely examine 0.0002% of the whole liver[5]. Moreover, they have a poor inter-reader agreement[4]. These limitations hamper the adequate management of patients with CLD−particularly so of those with NAFLD. Indeed, by contrast to antigenic or replication markers in viral

hepatitis[6], we lack biomarkers to assess the risk of liver disease progression in the tremendous NAFLD population[7]. Our current armamentarium is limited to blood- or imaging-based tools that focus on fibrosis evaluation−yet poorly perform in non-advanced fibrotic stages[7,8]. To date, we cannot reliably identify patients with a high degree of liver inflammatory activity ahead of the development of severe fibrosis. This undermines the possibilities for early management, such as close monitoring and/or therapeutic intervention(s). Likewise, we cannot evaluate non-invasively liver inflammation in current clinical trials and must rely on paired-biopsies that suffers from the hitherto limitations[9].

The hallmark of chronic liver inflammation is the infiltration of lymphocytes and macrophages in the liver parenchyma. While these cell populations are present in normal conditions[10], they rapidly

[1]Laboratory of Hepato-Gastroenterology, Institut de Recherche Expérimentale et Clinique, UCLouvain, Brussels, Belgium. [2]Department of Imaging and Pathology, KU Leuven, Leuven, Belgium. [3]Univ. Grenoble Alpes, INSERM, LRB, 38000 Grenoble, France. [4]Physiogenex, 31670 Labège, France. [5]Department of Medical Imaging, Laboratory of in vivo Cellular and Molecular Imaging, Vrije Universiteit Brussel, Brussels, Belgium. [6]Biocellvia, Marseille, France. [7]These authors contributed equally: Maxime Nachit, Christopher Montemagno. [8]These authors jointly supervised this work: Isabelle A. Leclercq, Pascale Perret. ✉e-mail: alexis.broisat@inserm.fr

expand through the recruitment of circulating leukocytes in response to cellular damage and proinflammatory signals[11]. At the molecular level, this recruitment is driven by attracting molecules and chemokines such as monocyte chemoattractant protein MCP-1[12]. In a coordinated manner, the adhesion proteins amongst which ICAM-1, VAP-1, and VCAM-1 promote the adhesion of circulating leukocytes to the endothelium prior to their subsequent migration to the liver[11,13–16]. Interestingly, the expression of VCAM-1 is regulated by the NF-κB program, a key initiator and mediator of liver inflammation in the context of non-alcoholic steatohepatitis (NASH), the progressive form of NAFLD[17,18]. Moreover, VCAM-1 was recently deemed to be a top upregulated gene and a key pathophysiological player in NASH[19].

Hence, based on mechanistic evidences supporting the implication of VCAM-1 in NAFLD pathophysiology[19] and given the profuse vascularization of the liver (thus the large scope for antibody-target binding), we speculated that molecular imaging of VCAM-1 could be a suitable non-invasive mean to detect liver inflammation ahead of fibrosis development. To test this hypothesis, we employed an anti-VCAM-1 nanobody labeled with technetium-99m ($^{99m}$Tc-cAbVCAM1-5, currently undergoing clinical evaluation for atherosclerosis imaging) in various preclinical models of NAFLD. This imaging agent is a cross-reactive binder of murine and human VCAM-1 proteins with nanomolar affinity as determined by surface plasmon resonance (KD = $2.0 \pm 0.0$ nM for murine and $6.5 \pm 0.7$ nM for human VCAM-1). $^{99m}$Tc-cAbVCAM1-5 was found to be well tolerated, stable in vivo in mouse blood up to 3 h, and exclusively eliminated through the kidneys−further supporting its use for liver imaging[20]. Therefore, we first performed a proof-of-concept study in mice fed a methionine and choline deficient (MCD) diet to evaluate $^{99m}$Tc-cAbVCAM1-5 specificity and sensibility for detecting liver parenchymal inflammation. This animal model was selected as MCD diet induces liver steatosis and significant parenchymal inflammation that can be easily reversed by switching to a normal diet[21]. We then employed four clinically-relevant models of NAFLD: high fat diet-fed wild-type and FOZ mice (WH and FH) and choline-deficient high fat diet-fed C57BL6/J (CDH) and choline-supplemented high fat diet-fed C57BL6/J (CSH) as to model variable degree of liver steatosis and parenchymal inflammation, with no or mild fibrosis. We quantified the signal emitted by the nanobodies using Single Photon Emission Computed Tomography (SPECT) and evaluated its diagnostic performance to detect liver parenchymal inflammation.

## Results

### Proof-of-concept study

Mice fed a MCD-diet lost ~40% of their body weight ($p < 0.0001$ versus those fed a standard diet; STD) and ~30% of their liver weight ($p < 0.0001$) (Fig. 1a, b). Of note, mice data are displayed separately according to their injection group in the imaging experiment (prefix "V-" for $^{99m}$Tc-cAbVCAM1-5 and "C-" for control nanobody, see below). Liver injury was supported by a 10-fold increase in plasma ALT levels in MCD-fed mice compared to STD (Fig. 1c). At the molecular level, the expression of the pro-inflammatory marker MCP-1 in the liver was increased 16-fold in MCD-fed mice when compared to STD ($p = 0.0015$) (Fig. 1d). Likewise, VCAM-1 mRNA expression in the liver was increased 6-fold in MCD-fed mice when compared to STD ($p = 0.0001$) (Fig. 1e). We evaluated mice livers in vivo using $^{99m}$Tc-cAbVCAM1-5 SPECT-CT imaging (Fig. 1f). In the V-MCD group, the liver uptake was significantly higher at 4 and 8 weeks in comparison to baseline and to control groups (Fig. 1f, g, $p < 0.001$). The biodistribution as evaluated ex vivo was well-correlated to SPECT quantification ($r^2 = 0.84$, $p < 0.0001$, Supplementary Fig. 1a) and further confirmed the ~2-fold increase in liver uptake in V-MCD versus V-STD mice ($0.41 \pm 0.14$ vs $0.26 \pm 0.03$ SUV, $p < 0.01$) (Supplementary Table 1). Intraclass correlation coefficient (ICC) for SPECT quantification was excellent (0.9694, Supplementary Fig. 1b). We then sought to determine whether the resolution of parenchymal inflammation would result in reduced $^{99m}$Tc-cAbVCAM1-5

liver uptake. Returning mice previously fed a MCD diet to a standard chow (V-Reverse) resulted in amelioration of liver fat, trend towards reduction of MCP-1, and almost complete normalization of F4/80 and VCAM-1 mRNA expression (Fig. 2a–d). We imaged mice at two time points: one week (W1) and four weeks (W4) after either STD-diet feeding (V-STD), MCD-diet feeding (V-MCD) or MCD-diet feeding and reversal to chow diet (V-Reverse) (Fig. 2e, f). In line with above results, $^{99m}$Tc-cAbVCAM1-5 liver uptake was higher in MCD-fed groups when compared to V-STD at W1 (Fig. 2e, f). At W4, $^{99m}$Tc-cAbVCAM1-5 liver uptake further increased in MCD-fed mice, while it significantly decreased in each of the animals of the Reverse group (Fig. 2f), thereby demonstrating the high sensitivity of the method. These proof-of-concept data support that cAbVCAM1-5 SPECT-CT imaging is suitable to detect liver inflammation in MCD-fed mice.

### Main study

**Characteristics of the NAFLD models.** Supplementary Table 2 summarizes preclinical models characteristics. High fat feeding in WT and FOZ (WH and FH) resulted in a higher body weight when compared to WT mice fed a normal diet (Ctl) ($54.7 \pm 4.7$ and $34.9 \pm 3.9$ g vs $29.8 \pm 2.3$ g) (Fig. 3a). As reported[21], mice fed a choline-deficient diet (CDH), despite a high fat content, had lower body weight when compared to siblings fed an identical diet supplemented with choline (CSH) ($28.6 \pm 3.3$ vs $21.2 \pm 1.5$ g) (Fig. 3a). Liver weight was much higher in FH when compared to WH and Ctl ($6.2 \pm 1.5$ vs $1.5 \pm 0.2$ g and $1.4 \pm 0.3$ g), and more modestly albeit significantly higher in CDH when compared to CSH ($1.6 \pm 0.2$ vs $1.1 \pm 0.1$ g) (Fig. 3b). Liver steatosis varied amongst groups: it was absent in Ctl, affecting less than 30% of hepatocytes in WH and CSH (most animals had a steatosis score of 1), and panlobular in FH and CDH (all animals had a steatosis score of 3) (Fig. 3c). Accordingly, steatosis area was $0.33 \pm 0.06\%$ in Ctl, $1.72 \pm 1.18\%$ and $0.87 \pm 0.55\%$ in WH and CSH, and as high as $17.37 \pm 4.62\%$ and $27.35 \pm 1.89\%$ in FH and CDH (Supplementary Fig. 2a, b). Hence, 22 out of the 32 mice of the total cohort had some degree of liver steatosis (0/8 Ctl, 5/6 WH, 6/6 FH, 5/6 CSH, and 6/6 CDH) (Fig. 3c).

**FH and CDH have panlobular macrovesicular steatosis and severe lobular inflammation.** While absent in Ctl, WH and CSH mice had mild steatosis and mild inflammation (>0 - <2 foci per 200× field, corresponding to 1 point in NAS score) (Fig. 3d). By contrast, FH and CDH mice with panlobular steatosis had severe inflammation (Fig. 3d). This gradation in inflammation is also reflected by the number of crown-like structures (CLS) per area, representing aggregated macrophages around steatotic hepatocytes, that was higher in FH when compared to WH and Ctl ($37.0 \pm 7.7$/mm² vs $6.9 \pm 8$/mm² and $1.8 \pm 2.6$/mm², respectively) (Supplementary Fig. 2c), and even much more numerous in CDH ($133.8 \pm 33.4$/mm²), while rarely present in CSH livers ($2.2 \pm 1.6$/mm²) (Supplementary Fig. 2d). FH had severe ballooning (Fig. 3e) and NAFLD activity score (NAS) was 0 in Ctl, ≤4 in WH and CSH and ≥6 in FH and CSH (Fig. 3f). Representative histological sections are displayed in Fig. 3g. In keeping with histological findings, we found a higher expression of F4/80 and MCP-1 in FH when compared to WH (F4/80: $5.29 \pm 1.44$ vs $1.32 \pm 0.58$ fold, and MCP-1: $7.71 \pm 1.15$ vs $1.15 \pm 0.6$ fold) and in CDH when compared to CSH (F4/80: $6.66 \pm 3.21$ vs $2.57 \pm 0.84$ fold, and MCP-1: $12.51 \pm 2.99$ vs $1.45 \pm 0.80$ fold) (Fig. 4a, b). Thus, based on biological data and liver histology, FH were overtly obese, FH and CDH had panlobular steatosis with severe lobular inflammation (with a much higher liver mass in FH), WH and CSH exhibited a spectrum of liver phenotypes ranging from normal to simple steatosis with or without mild inflammation, and Ctl had normal liver. Mild pericellular fibrosis developed in FH and CDH livers (Supplementary Fig. 3a, b).

**VCAM1 expression correlates with inflammatory markers.** VCAM-1 mRNA expression was significantly higher in FH and CDH than in their respective controls (Fig. 4c), a pattern reminiscent of that of F4/80 or

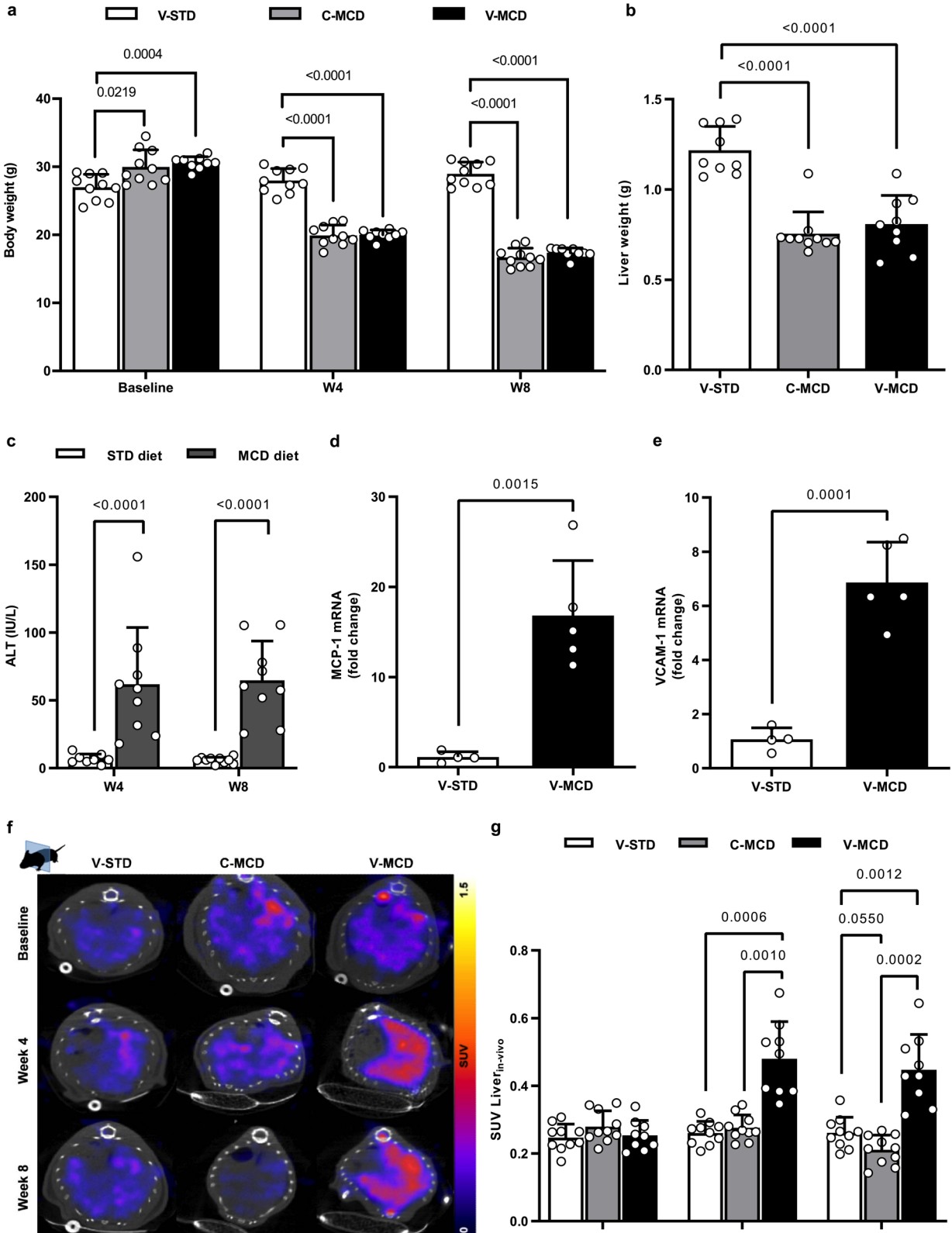

**Fig. 1 | Specificity of $^{99m}$Tc-cAbVCAM1-5 SPECT-CT imaging. a** Body weight ($n = 10$/group/time point, except V-MCD group ($n = 9$); two-way ANOVA), **b** liver weight ($n = 10$/group, except V-MCD group ($n = 9$); one-way ANOVA), **c** plasma ALT activity (STD-fed ($n = 10$) and MCD-fed ($n = 9$) mice; unpaired Student's t-test), **d** liver MCP-1 mRNA (V-STD ($n = 4$) and V-MCD ($n = 5$) group; unpaired Student's t-test) and **e** liver VCAM-1 mRNA (V-STD ($n = 4$) and V-MCD ($n = 5$) group; unpaired Student's t-test) in STD-fed mice injected with $^{99m}$Tc-cAbVCAM1-5 (V-STD), MCD-fed mice injected with control nanobody (C-MCD) and MCD-fed mice injected with $^{99m}$Tc-cAbVCAM1-5 (V-MCD). **f** Representative SPECT-CT liver imaging with $^{99m}$Tc-cAbVCAM1-5 or $^{99m}$Tc-Control at baseline, 4 and 8 weeks; transversal sections and **g** SPECT quantification in SUV (C-MCD & V-STD mice ($n = 10$), and V-MCD mice ($n = 9$); two-way ANOVA). All data are represented as mean ± SD and significant $p$ values (corrected for multiple testing using Tukey's post hoc when more than two groups were compared) are given above the bars. Source data are provided as a Source Data file.

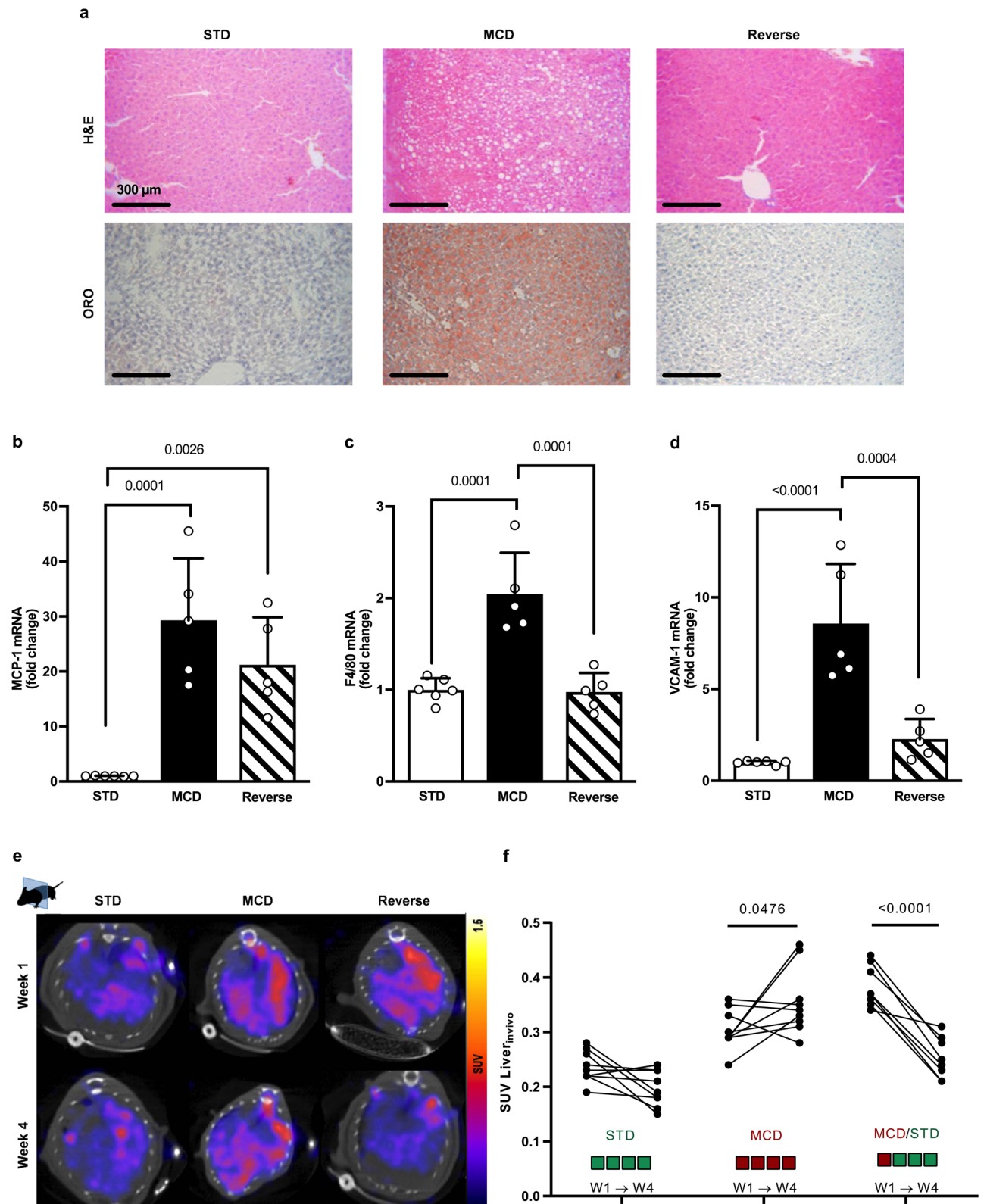

**Fig. 2 | Liver uptake of $^{99m}$Tc-cAbVCAM1-5 upon steatohepatitis reversal.**
**a** Representative Hematoxylin & Eosin (H&E) and Oil red-O (ORO) liver staining (x200) in all groups. Liver mRNA expression of **b** MCP-1, **c** F4/80, and **d** VCAM-1 in the three groups of mice (one-way ANOVA, $n = 6$ for V-STD and $n = 5$ for V-MCD & V-Reverse groups, corrected for multiple comparisons using Tukey's post hoc).

**e** Representative SPECT-CT liver imaging with $^{99m}$Tc-cAbVCAM1-5 at 1 and 4 weeks, transversal sections. **f** SPECT quantification in SUV (two-way ANOVA, $n = 9$ for V-STD & V-MCD mice and $n = 8$ for V-Reverse, corrected for multiple comparisons using Šídák's test). All data are represented as mean ± SD and significant $p$ values are given above the bars. Source data are provided as a Source Data file.

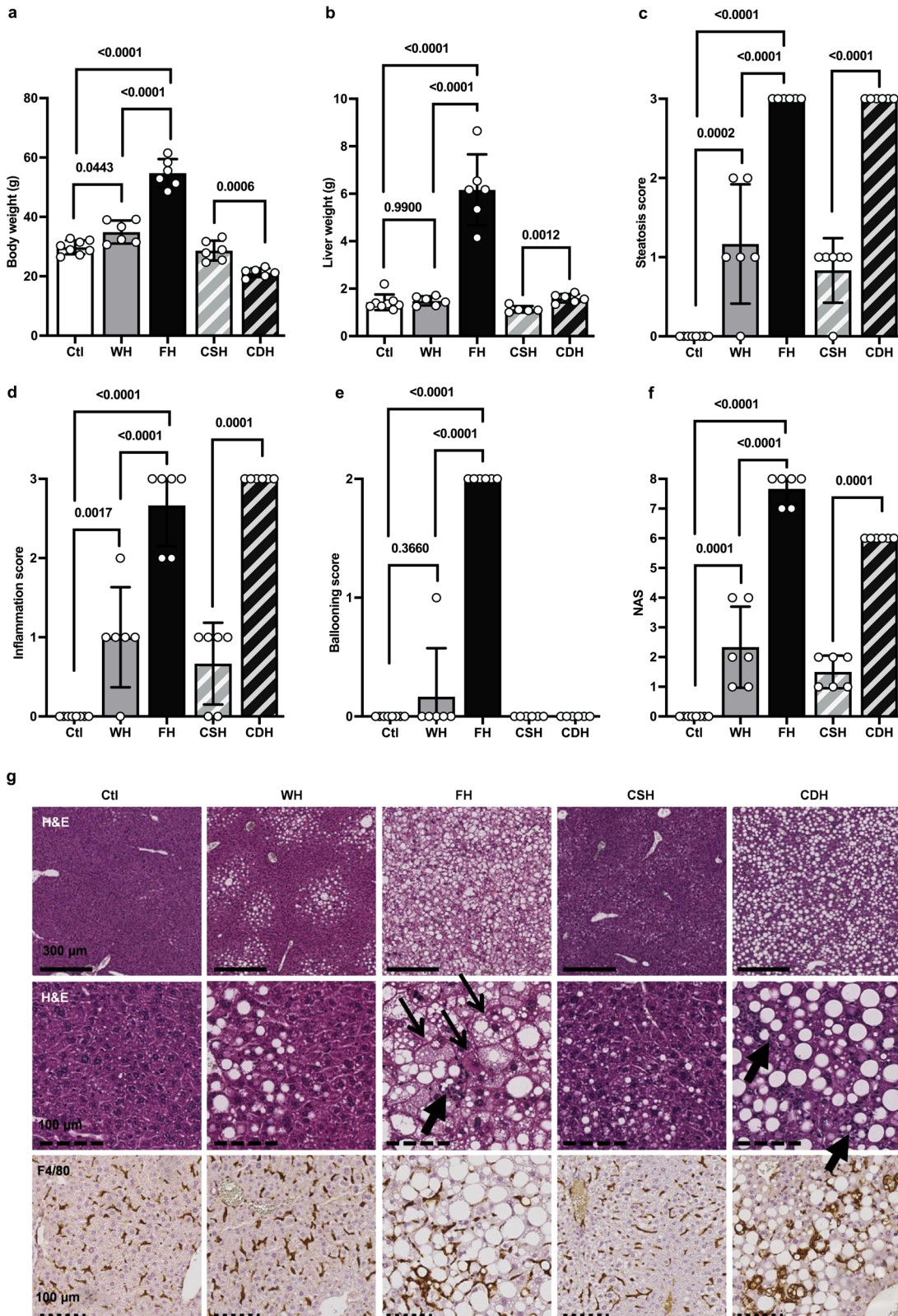

**Fig. 3 | Characteristics of the NAFLD models. a** Body weight, **b** liver weight, and histological **c** steatosis score, **d** inflammatory score, **e** ballooning score, and **f** total NAS score in WT mice fed a normal diet (Ctl, $n = 8$) or a high fat diet (WH, $n = 6$), in FOZ mice fed a high fat diet (FH, $n = 6$) and in C57BL6/J mice fed a fat-rich choline supplemented (CSH, $n = 6$) or deficient (CDH, $n = 6$) diet (one-way ANOVA and Welch's t-test). **g** Representative liver histology in all groups (thick arrow = inflammatory loci, thin arrow = ballooning). All data are represented as mean ± SD and significant $p$ values (corrected for multiple testing using Tukey's post-hoc when more than 2 groups were compared) are given above the bars. Source data are provided as a Source Data file.

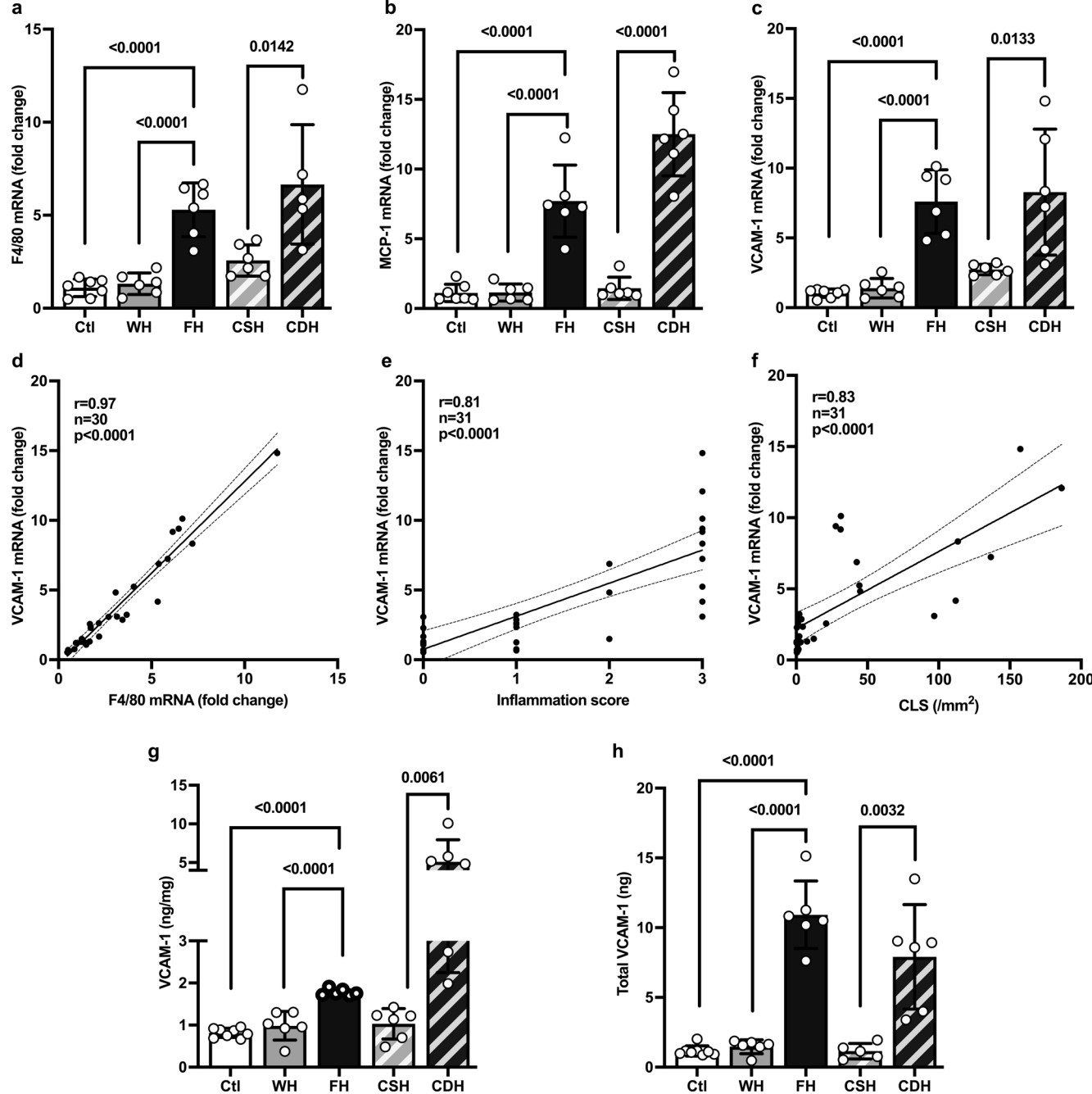

**Fig. 4 | VCAM1 expression correlates with inflammatory markers. a** F4/80, **b** MCP-1, and **c** VCAM-1 mRNA fold change (one-way ANOVA and Student's t-test, $n = 7$ for Ctl and $n = 6$ for other groups). Spearman's correlations between VCAM-1 mRNA level (fold change compared to housekeeping gene RPL19) and **d** F4/80 fold change, **e** histological inflammatory score, and **f** CLS, respectively. **g** VCAM-1 protein content per mg of tissue (one-way ANOVA and Student's t-test, $n = 8$ for Ctl and $n = 6$ for other groups). **h** Total ng of VCAM-1 in the liver (one-way ANOVA and Student's t-test, $n = 8$ for Ctl and $n = 6$ for other groups). All data are represented as mean ± SD and significant $p$ values (corrected for multiple testing using Tukey's post-hoc when more than 2 groups were compared) are given above the bars. Source data are provided as a Source Data file.

MCP-1 (Fig. 4a, b). When data from each individual animal were considered, VCAM-1 expression strongly correlated with F4/80, histological inflammatory score and CLS density ($r = 0.97$, $r = 0.81$, and $r = 0.83$ respectively, all $p < 0.0001$) (Fig. 4d–f). At the protein level, high VCAM-1 content was found in FH and CDH livers (Fig. 4g). The VCAM-1 protein content per mg of liver tissue was higher in FH and CDH than in the other groups, although it was much higher in CDH than in FH (Fig. 4g). We also calculated the VCAM-1 protein content in the entire liver (Fig. 4h). The total VCAM-1 load was found much higher in FH and CDH livers than in the liver of control animals.

**$^{99m}$Tc-cAbVCAM1-5 liver uptake correlates with VCAM-1 expression.** Representative SPECT-CT liver imaging with $^{99m}$Tc-cAbVCAM1-5 are displayed in Fig. 5a for all preclinical models. Both in vivo (Fig. 5b) and ex vivo (Fig. 5c), the SUV was significantly higher in WH and FH when compared to Ctl, and in CDH when compared to CSH (Fig. 5b, c). There was a tight correlation between ex vivo and in vivo measurements ($r = 0.93$, $p < 0.0001$) (Supplementary Fig. 4). The biodistribution was evaluated ex vivo and is available in Supplementary Tables 3 and 4. To evaluate the specificity of the nanobody binding, we co-injected an additional group of CDH with $^{99m}$Tc-cAbVCAM1-5 and with

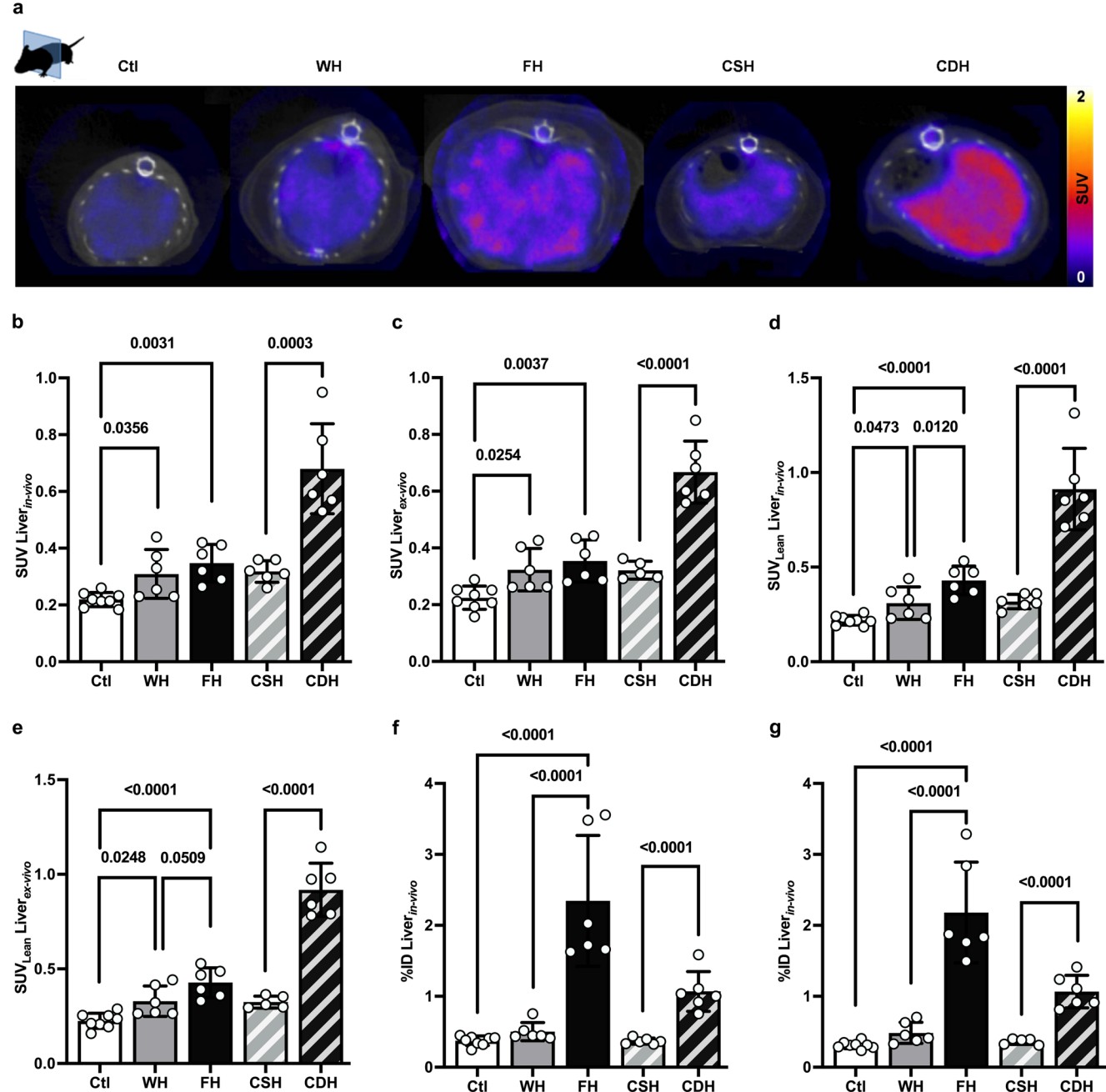

**Fig. 5 | $^{99m}$Tc-cAbVCAM1-5 liver uptake correlates with VCAM-1 expression.** SUV Liver measured in all groups **a** in vivo by SPECT or **b** ex vivo (one-way ANOVA and Student's t-test, $n = 8$ for Ctl and $n = 6$ for other groups). **c** Representative SPECT transversal images. SUV$_{Lean}$ Liver **d** in vivo and **e** ex vivo, and total liver uptake (%ID) **f** in vivo and **g** ex vivo of $^{99m}$Tc-cAbVCAM1-5 evaluated with SPECT (one-way ANOVA and Student's t-test, $n = 8$ for Ctl and $n = 6$ for other groups). All data are represented as mean ± SD and significant $p$ values (corrected for multiple testing using Tukey's post-hoc when more than 2 groups were compared) are given above the bars. Source data are provided as a Source Data file.

a 100x excess cold nanobody (Supplementary Fig. 5). SUV was of 0.89 ± 0.10 in the group without competitor and the competition resulted in a significant decrease to 0.21 ± 0.03 ($p = 0.0078$), a value similar to that previously observed in control animals or when using an irrelevant nanobody. Expectedly as in the models used fibrosis was mild, if any, we found no correlations between VCAM-1 content and fibrosis (Supplementary Fig. 6a). Likewise, there was no or weak correlations between VCAM-1 signal and fibrosis (Supplementary Fig. 6b–d). Multiple linear regression analysis confirmed that inflammation (and to a lesser extent ballooning), but not fibrosis, were independent histological predictors of VCAM-1 signal (Supplementary Table 5).

**$^{99m}$Tc-cAbVCAM1-5 uptake robustly reflects on liver inflammation in a fatty liver.** As elaborated above, SUVs were higher in FH and CDH when compared to their respective controls, reflecting on severe liver inflammation in these models. Accordingly, $^{99m}$Tc-cAbVCAM1-5 signal quantification had a strong diagnostic power to detect any degree of inflammation (histological score [1–3]) or moderate-to-severe inflammation (histological score [2-3]), returning respective areas under the curves (AUCs) of 0.93 and 0.91 ($n = 32$, all $p < 0.0001$) when all animals were considered (Ctl, WH, FH, CSH, and CDH) (Supplementary Fig. 7a). The diagnostic power was attenuated when considering only animal with NAFLD (Supplementary Fig. 7b). However, FH had a much higher liver weight and steatosis degree than WH (Fig. 3b). Therefore, in the

context of steatotic hepatomegaly, we hypothesized that $SUV_{Lean}$ Liver (i.e., SUV corrected for steatosis degree, see Methods and Supplementary Fig. 8) and %ID Liver (i.e., total liver uptake) could better witness the inflammatory recruitment capacity of the liver than crude SUV. $SUV_{Lean}$ Liver and %ID Liver were strongly correlated to histological inflammatory scores ($r = 0.89$ and $0.82$, respectively, all $p < 0.0001$) (Supplementary Fig. 9a, b). Whether measured ex vivo or in vivo, $SUV_{Lean}$ Liver (Fig. 5d, e) and %ID Liver (Fig. 5f, g) were higher in FH than in WH or Ctl. Likewise, $SUV_{Lean}$ Liver and %ID Liver were higher in CDH than in CSH livers (Fig. 5d–g). Hence both approaches better discriminated between WH and FH groups. When $^{99m}$Tc-cAbVCAM1-5 signal quantification was based on $SUV_{Lean}$ Liver or %ID Liver, the diagnostic power to detect any degree of inflammation or moderate-to-severe inflammation was excellent (AUCs 0.86–0.99, all p < 0.05) (Fig. 6a–d). Strikingly, AUCs based on $SUV_{Lean}$ Liver or %ID Liver were consistently ≥ 0.95 to distinguish moderate-to-severe from no or mild liver inflammation, whether in the whole cohort or when only animals with NAFLD were considered. Hence, SPECT imaging with $^{99m}$Tc-cAbVCAM1-5 is a robust method to detect liver inflammation in NAFLD and can be fine-tuned based on liver steatosis degree and/or morphometrical changes at the individual level.

## Discussion

Because patients with CLD (and particularly those with NAFLD) are so numerous, it is unconceivable to propose a liver biopsy as a screening method for liver inflammation[7]. Yet, patients with a high degree liver inflammation are at risk to progress to severe liver disease including cirrhosis and hepatocellular carcinoma[22] as well as facing extra-hepatic complications, in particular cardiovascular events[23,24]. In the present study, we repurposed the anti-VCAM-1 nanobody-based tracer $^{99m}$Tc-cAbVCAM1-5 (originally developed for the detection of inflammatory atherosclerotic lesions[20,25]) to non-invasively evaluate liver inflammation in preclinical models of NAFLD. We first confirmed that VCAM-1 expression, at the transcriptomic level as well as at the protein level, strongly reflects on histological and molecular markers of liver inflammation. This paradigm held true irrespectively from the mice genetic background, the diet, the dysmetabolic context, or the severity of liver steatosis. We next demonstrated that the signal generated by $^{99m}$Tc-cAbVCAM1-5, measured in vivo and validated ex vivo, highly correlates with VCAM-1 protein content in the liver. The proof-of-concept study illustrated that the amelioration of parenchymal inflammation obtained by interrupting the MCD diet was associated with a significant decrease of $^{99m}$Tc-cAbVCAM1-5 liver uptake, confirming the sensitivity of the method. Finally, receiving operating curves (ROC) returned a remarkable diagnostic performance for the non-invasive detection of any degree of liver inflammation in multiple preclinical NAFLD models (AUC 0.85–0.99). These pilot data support that VCAM-1 molecular imaging could be a first-in-class tool to identify amongst subjects with apparently benign liver disease (i.e., no or mild fibrosis), those with severe liver inflammation—hence the most at-risk for progression.

Besides, we evaluated alternative signal quantification methods to the crude SUVs as we suspected that in the context of NAFLD, the morphological changes due to severe steatosis (with or without hepatomegaly, as found in FH and CDH respectively) might artificially decrease VCAM-1 density (content per volume of lean tissue). This possibility was fully supported by the differential results yielded by the normalization of VCAM-1 content per mg of tissue or for the entire organ. We tested two approaches: the $SUV_{Lean}$ Liver and the %ID Liver (see "Methods"). Supporting our hypothesis, both $SUV_{Lean}$ Liver and %ID Liver had a higher diagnostic performance than SUV to detect any degree of liver inflammation (1–3 histological score) or moderate-to-severe liver inflammation (2–3 histological score), particularly so when only mice with NAFLD were considered. Clinical studies will be required to compare these methods (i.e., SUV, $SUV_{Lean}$, and %ID) and nominate

the best-performing one in a real-life scenario. Such examination could be regularly proposed to patients that are particularly susceptible to progressive liver disease, e.g., those with type 2 diabetes (T2D)[26] or with particular genetic polymorphism such as PNPLA3[27]. Taken together, we foresee that VCAM-1 imaging could be a key asset in the hepatologist's toolbox to guide the management of patients with NAFLD and potentially other CLDs. The same reasoning applies for the evaluation of treatment efficacy in clinical trials. At the present time, intermediate analyses (e.g., before a second biopsy) are often limited to the highly imperfect use of steatosis reduction as a surrogate marker for liver disease improvement[28]. Methodologies to track liver inflammation status (and potential regression) are thus eagerly awaited. We speculate that $^{99m}$Tc-cAbVCAM1-5 imaging could be used to rapidly identify (non)-responders, enabling tailored therapeutic strategies.

Few other radiotracers have been evaluated so far for the imaging of liver inflammation. Some target $\alpha_v\beta_3$ expressing stellate cells[29] or type I collagen[30] for the identification of early or late fibrosis. Radiotracers targeting at liver metabolism, including mitochondria, have also been evaluated, often leading to a decrease in liver uptake during NASH[31]. Inflammation imaging has been evaluated using nanobody derived radiotracers targeting at Kupffer cells in a mouse MCD model of NASH[32]. A key advantage of $^{99m}$Tc-cAbVCAM1-5 over the above-mentioned radiotracers is that $^{99m}$Tc-cAbVCAM1-5 is not taken up by the healthy liver. Indeed, the expression of the target protein (VCAM-1) is part of the inflammatory response and is likely less influenced by resident cell populations (e.g., Kupffer cell). However, we must acknowledge the possibility that in end-stage fibrosis or cirrhosis, the VCAM-1 signal could be distorted, as a consequence of reduced accession of the nanobody to the sinusoidal endothelium and transformation (capillarization) of the endothelium. This question was not addressed in our study because there are efficient (point of care) noninvasive tools to identify end-stage liver disease, e.g., elastography or blood-based scores[8].

Taken collectively, given that VCAM-1 is a key player in NASH pathogenesis[19], that its liver expression in mice and human is upregulated with NASH (and maintained with fibrosis progression)[19] and that the $^{99m}$Tc-cAbVCAM1-5 employed in the present study is already in phase 1 clinical trial (NCT04483167) for atherosclerotic lesion imaging, the coherent follow-up of our results is to test $^{99m}$Tc-cAbVCAM1-5 imaging to evaluate liver inflammation in patients with NAFLD, a currently unmet medical need.

## Methods
### Animal models

All procedures were approved by the animal care and ethic committee of Grenoble Alpes University and the ad hoc French minister as well as by Belgian institutions (APAFIS#2993-2015120219565475 & APAFIS#23780-2020012412159346_v2 and 2016/UCL/MD/003). A one-week acclimatization period was respected prior diet onset and, when applicable, randomization was performed so that mean weights were similar between groups. Mice had access to rodent chow and water ad libitum. Mice were on a 12 h light/dark cycle, temperature was maintained between 20 and 24 °C, and a relative humidity of 40–60%.

*Proof-of-concept study*: Thirty 12 weeks-old male C57Bl/6J (Charles River) were used to investigate the kinetic and specificity of SPECT imaging. $^{99m}$Tc-cAbVCAM1-5 liver uptake was investigated at baseline, 4 weeks, and 8 weeks in mice fed a MCD diet (V-MCD) in comparison to mice fed a standard diet (V-STD), and to an irrelevant control nanobody (C-MCD) ($n = 10$ per group; one mouse was excluded in V-MCD group since a limit point concerning the body weight was reached at W8 under MCD diet). Then the sensitivity was evaluated: SPECT imaging was performed at week 1 and week 4 in mice fed a MCD diet for one week followed by 3 weeks of reversal to standard diet (Reverse), in comparison to mice fed a chow (STD) or a MCD diet for 4 weeks ($n = 9$ per group).

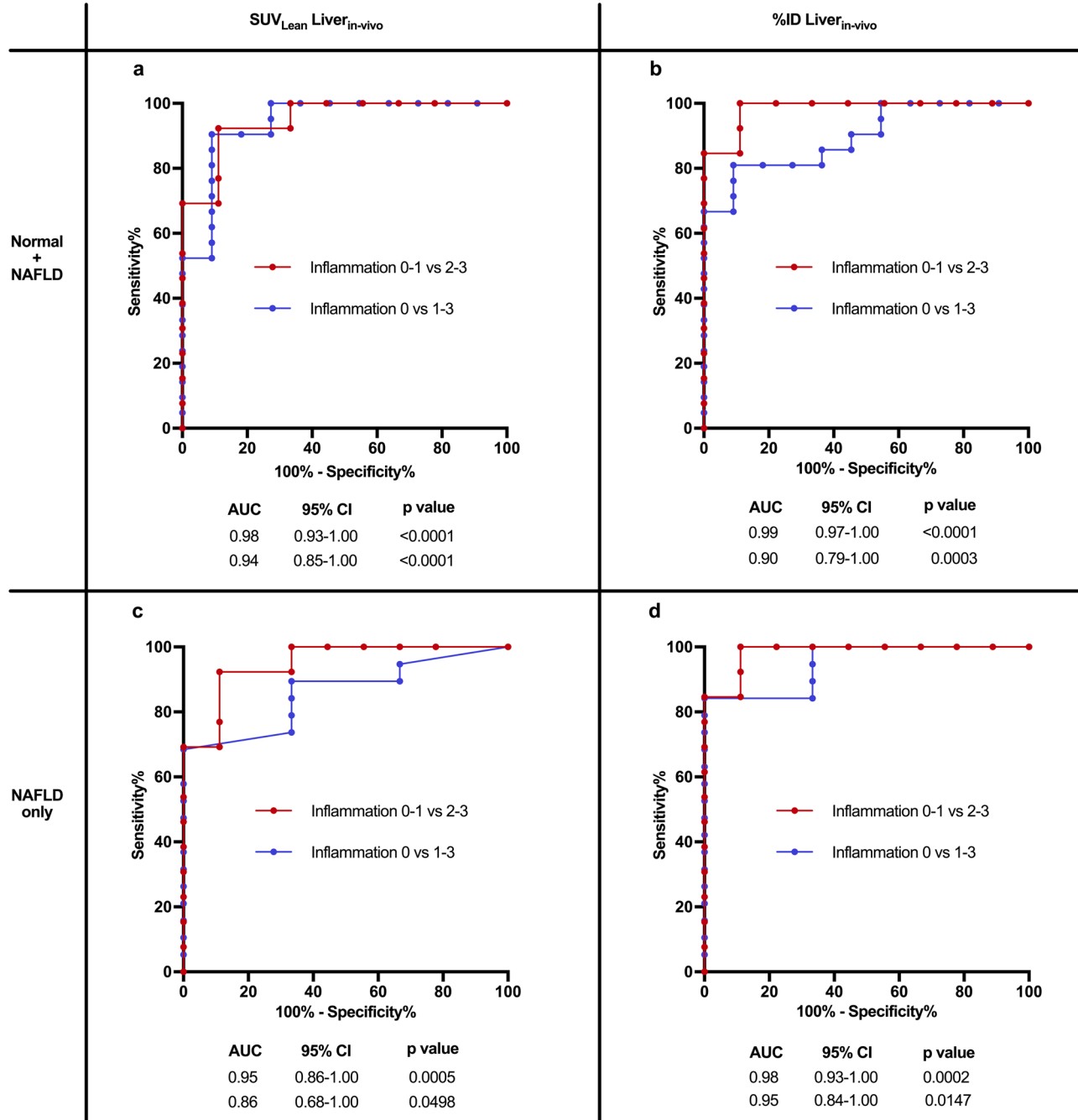

**Fig. 6 | 99mTc-cAbVCAM1-5 uptake robustly reflects on liver inflammation in a fatty liver.** ROC curves for detection of liver inflammation **a** in the entire cohort (Normal + NAFLD) using $SUV_{Lean}Liver_{in-vivo}$ values, or **b** using %ID $Liver_{in-vivo}$ values, and **c** only in animals with liver steatosis (NAFLD only) using $SUV_{Lean}Liver_{in-vivo}$ values, or **d** using %ID $Liver_{in-vivo}$ values. AUC: area under the curve calculated using Wilson/Brown method, 95% confidence interval (CI) and *p* values are provided on each panel. Source data are provided as a Source Data file.

*Main study*: Twelve male C57Bl/6J mice (Charles River) were fed a choline-deficient high fat diet (CDH, *n* = 6) or a choline-supplemented high fat diet (CSH, *n* = 6) for 4 weeks. The FOZ mouse strain on a non-obese diabetic B10 background[33] was also studied as an alternative model of NAFLD, highly resembling the phenotype seen in humans (i.e., obesity, insulin resistance, and NAFLD). Heterozygous mice were used for breeding, and homozygous FOZ (*n* = 6) and wild type WT (*n* = 14) male littermates for the dietary experiments. FOZ and WT mice received a high fat diet for 12 weeks (FH, *n* = 6; WH, *n* = 6) as models of NAFLD. WT mice fed a standard rodent chow served as controls (Ctl, *n* = 8).

## cAbVCAM1-5 and Control nanobodies radiolabeling

The cAbVCAM1-5 and nonspecific Control (cAbBcII10) nanobodies were produced and radiolabeled as previously described using the tricarbonyl method[20]. Briefly, $[^{99m}Tc(H_2O)_3(CO)_3]^+$ ($^{99m}Tc$-tricarbonyl) was synthesized by adding 1 mL of $^{99m}TcO_4^-$ solution to an Isolink kit (PSI). The vial was incubated at 100 °C for 20 min and neutralized with HCl. Then, 0.5 mL of $^{99m}Tc$-tricarbonyl was added to 50 μg of cAbVCAM1-5 or cAbBcII10 and 50 μL of 50 mM carbonate buffer, pH 8.0. This mixture was incubated for 45 min at 75 °C for cAbVCAM1-5, and for 90 min at 50 °C for cAbBcII10. Each solution was further gel-filtrated on Sephadex G25 (NAP-5; GE Healthcare) and filtered (0.22 μm

Millex; Millipore). Radiochemical purity, determined by reverse-phase high-performance liquid chromatography, was more than 98% for both radiotracers.

## cAbVCAM1-5 and competition experiment

To evaluate the specificity of the cAbVCAM1-5 binding, [99m]Tc-cAbVCAM1-5 (5 μg) was either injected alone or in combination with a 100× excess of cold cAbVCAM1-5 (500 μg) in an additional group of C57Bl/6J mice fed a choline-deficient high fat diet (CDH) for 4 weeks (*n* = 5 per group).

## SPECT-CT imaging

SPECT-CT acquisitions were performed 1 h after the intravenous injection of [99m]Tc-cAbVCAM1-5 or [99m]Tc-Control (-37 MBq) using the nanoscan SPECT-CT from Mediso as previously described[20]. Volumes of Interest (VOI) were drawn using the axial view CT images (Vivoquant, Invicro), blinded from the SPECT. Radioactivity was expressed as a concentration: Standardized Uptake Value (SUV) = liver activity (MBq/g) / injected dose (MBq/g) and as a total liver uptake (%Injected Dose/organ; %ID/organ). Inter-observers' reproducibility was evaluated. Of note, %ID/organ (referred to as %ID Liver hereafter) and SUV do not directly correct for liver steatosis. Given that liver fat is not participating to signal generation (and represents up to 30% of dead volume), we posited that the correction of the VCAM-1 signal by the proportion of lean liver might better reflect on the inflammatory status of the organ. We thus developed a SUV$_{Lean}$ Liver (reminiscent of the SUV$_{Lean}$ wherein the SUV is corrected by lean body mass), based on histological steatosis % (ex vivo) or CT-based Hounsfield Unit (HU) (in-vivo). To compute the latter, we first verified that liver density in HU was correlated with histological steatosis. We found a high correlation, especially for steatosis > 5% (i.e., the definition of NAFLD) (Supplementary Fig. 8). Given that CT HU values exhibit a large variability when steatosis <5% (as expected from the human literature[34]) – and that such cases would be rare in our target NAFLD population—we corrected for steatosis when liver density was <−50 HU (-5% steatosis) (Supplementary Fig. 8) and derived the following formula: liver steatosis (%) = −0.2476*Liver HU −3.945. We then divided the SUV Liver$_{in-vivo}$ by the lean liver % [i.e., 100% − CT-based liver steatosis (%)], yielding a SUV$_{Lean}$ Liver$_{in-vivo}$.

## Post-mortem analysis

2 h after radiotracer injection and immediately following SPECT-CT acquisition, anesthetized mice were euthanized using $CO_2$ and liver was harvested along with major organs. Samples were weighed and the radioactivity determined (Wizard[2] γ-counter, PerkinElmer). Results were corrected for decay, injected dose, organ weight, and expressed as SUV$_{ex-vivo}$.

## ELISA

Five (5) mg of liver was grounded in 100 μL RIPA buffer (50 mM Tris-HCl, 150 mM NaCl, 0.1% SDS, 0.5% sodium deoxycholate, 1 mM sodium orthovanadate, 1% protease inhibitor cocktail). Lysate was centrifuged at 10,000 × *g* for 10 min at 4 °C and supernatant was collected. VCAM-1 protein level was quantified using a commercial kit (Abcam, SimpleStep ELISA ab201278).

## ALT assay

Alanine Aminotransferase Activity (ALT) was assessed on the plasma of STD and MCD diet fed mice (at 4 and 8 weeks) with the ALT activity assay (Sigma-Aldrich), according to the manufacturer protocol. Results are express in International Unit (IU)/mL.

## Histology and Immunohistochemistry

*Proof-of-study:* Ten-micrometer frozen liver slices were colored with Oil-Red-O for lipids detection, *n* = 3–4 slices/liver. For immunohistochemistry, slices were incubated with a rat anti-mouse Mac-2 antibody (1:500) (CedarlaneLabs) to reveal activated macrophages, *n* = 3/liver.

*Main study:* Hematoxylin & Eosin (H&E) staining on 5 μm thick sections of paraffin-embedded livers was performed to assess the NAFLD activity score (NAS) according to Kleiner et al.[35], which is composed of semi-quantitative sub-scores ranging from 0-3 for steatosis, 0–2 for ballooning and 0–3 for lobular inflammation[36]. Automated digital analysis was used to measure steatosis (calculated based on the total area of lipid vesicles on the area of the liver section from H&E sections). For immunohistochemical detection of F4/80, slices were incubated with a rat anti-mouse F4/80 antibody (1/200) (MCA497G, BioRad). Crown-like structures (CLS), which are aggregation of active macrophages around fat-laden hepatocytes, were quantified on F4/80-stained liver sections[37]. Picrosirius red staining was performed to assess fibrosis. The fibrosis proportionate area (%) was measured using QuPath (v0.3.0) and a Random Forest-based pixel classifier. Histological analyses were performed on all mice in the Main study (*n* = 32).

## RT-qPCR assay

Extraction of total RNA from liver tissue (20 mg) was performed using PureLink RNA Mini Kit (ThermoFisher scientific) according to the manufacturer's protocol. Reverse transcription (RT) was performed using an iScript Reverse transcription kit (Biorad). Quantitative Polymerase Chain Reaction (qPCR) was performed on the resulting cDNA with the Fast SYBR Green Master Mix (Life technologies), using a Real Time PCR system (Applied Biosystem StepOne Plus; ThermoFisher Scientific). The expression levels of VCAM-1, F4/80 and MCP-1 mRNAs were normalized to the endogenous control actin, and were calculated with the formula 2-ΔΔCT. Primer sequences are given in Supplementary Table 6.

## Statistics

Data were expressed as mean ± standard deviation (SD). Statistical differences between groups were evaluated using two-sample t-test (2 groups) or one-way ANOVA (≥3 groups). For paired values at different time-points, two-way ANOVA was used. Of note, Ctl, WH and FH were analyzed with one-way ANOVA while CSH vs CDH were analyzed with Student's or Welch's (in case of unequal variance) t-test, but results from the two separate experiments were presented in a merged graph to facilitate interpretation. Significance of linear correlations was assessed with a Pearson's or Spearman's test. *P* values <0.05 were considered significant. Most analysis were conducted with GraphPad 9 (Prism) and R (v4.1). Intra-class correlation coefficients (ICC) were calculated using two-way random and absolute agreement (Medcalc 12.7).

## Reporting summary

Further information on research design is available in the Nature Portfolio Reporting Summary linked to this article.

## Data availability

All data presented in graphs within the Figures & Supplementary Figures are provided in the Source Data file. All images (IHC and SPECT/CT) are available from the corresponding author [AB]. Source data are provided with this paper.

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

## Acknowledgements

This work was partly supported by the Ph.D. fellowship from FRIA (FNRS, Belgium) [grant number 31618719 (to M.N.)], by the Fund for Scientific Medical Research (FNRS Belgium) [grant number T.0141.19 (to IAL)] and by France Life Imaging, grant ANR-11-INBS-0006 (to C.G.).

## Author contributions

M.N. and C.M. were involved in data analysis and interpretation, and manuscript drafting. M.A. and S.B. were involved in data acquisition, analysis and interpretation (radiochemistry), and manuscript drafting. R.C. was involved in data acquisition (biology) and manuscript reviewing. C.S. was involved in data analysis and interpretation (histology) and manuscript drafting. F.B., N.D., T.S., and C.G. were involved in conception and design (biology and radiochemistry) as well as in data analysis and manuscript reviewing. I.L., P.P., and A.B. were involved in study design, data acquisition, analysis and interpretation, and manuscript drafting. All authors gave final approval of the manuscript. M.N. is randomly listed before co-first author C.M.

## Competing interests

A. Broisat, C. Ghezzi and N. Devoogdt are inventors on the patent on VCAM-1 Nanobodies (PCT/EP2012/066348), granted amongst other countries in US and Europe (US9771423B2 and EP2748196B8). Other authors declare no competing interests.
