## [Peer Review File · Nature Communications]

REVIEWER COMMENTS

Reviewer #1 (Remarks to the Author):

In this study the authors determine if the molecular imaging von VCAM-1 could be used as an alternative non-invasive tool to detect liver inflammation in the setting of chronic liver disease. Key findings include that in various preclinical models of liver disease the uptake correlated with liver inflammation und molecular markers of inflammation. While the authors present very interesting data there are several issues regarding the study design and interpretation of the data that need to be addressed:

Proof of concept study: It´s not clear why MCP-1 was chosen. Additional marker like NAS and infiltrating neutrophils that are related to NAFLD and NASH should be presented in addition to bolster the conclusion that VCAM-1 might be a marker to detect inflammation. Also, how about AST? And VCAM protein or mRNA?

How is the detection affected by the presence of fibrotic alterations? Please show some data from models with later stages of the disease, preferable the same model (not MCD model) at different stages of the disease.

The representative pictures shown of the histology of livers are rather poor in quality. And while this is not the main focus of the study, better pictures would be very beneficial. Also please provide more characterization of the models used in regard to liver damage and inflammation.

How is VCAM-1 expression in different stages of NAFLD in humans?

Are there any adverse effects of the labeled VCAM-1?

Line 138: soften the statement: liver inflammation in an MCD-fed mouse.

Reviewer #2 (Remarks to the Author):

Nachit et al. Molecular imaging of liver inflammation using an anti-VCAM-1 nanobody. Nat Comm April 2022

General comments

New endpoints to assess and quantify liver inflammation is an urgent clinical need, given the increase in NAFL and diabetes worldwide. Thus, accurate diagnostics are welcome. Furthermore, novel interventions require sensitive, repeatable and non-invasive assessment of treatment effect – this is certainly lacking today which is a major obstacle for drug development and approval.

Nachit et al has repurposed a novel Nanobody binder to VCAM-1, with the hypothesis that non-invasive SPECT imaging of this protein would provide a suitable biomarker for liver inflammation. Nanobodies are small (≈ 15 kDa) proteins which theoretically are excellent scaffolds for imaging agents.

The study is well motivated and interesting but with major limitation and quite a lot of missing background information which makes it difficult to assess and interpret.

Major comments

INTRODUCTION

1) The nanobody 99Tcm-cAbVCAM1-5 is poorly described which makes it very difficult to interpret the results. What is the affinity to human VCAM-1? Murine VCAM-1. Seems to be \approx nM when searching the literature, but this should be explicitly stated. What is the stability in vivo and in plasma?

METHODS/ RESULTS.

2) The pilot and main studies are analysed using different endpoints (SUV and %ID respectively).

In the end %ID is recommended by the authors. The %ID should be calculated also for the pilot study, and correlated to other assessment of inflammation, like in the main study.

3) Its unclear why %ID would be a better endpoint than SUV (or %ID/g). SUV or %ID/g normally represents the concentration of the SPECT tracer or the target receptor – and I assume that we would like to measure the density of the liver inflammation, i.e. the “amount of inflammatory activity” per liver volume? %ID would measure the total amount of inflammation in a whole liver, but this would lead to potentially very strange results. Consider two persons, who both have a liver uptake of SUV = 1. Person 1 is tall with a large liver of 2.5 litres. Person 2 is a smaller person with a liver volume of 1.25 litres. Neither has NAFL. But %ID will be 2 times higher in person 1 – even though only the liver size is larger, not the concentration of inflammation.

I understand the attempt to correct for increased steatosis, but this must be done in a more advanced manner, for example by measuring steatosis by MR or similar.

4) Is there a reason why VCAM-1 wasn't stained for on sections? Other inflammation/ macrophage markers were used instead. The authors should comment and explain this decision.

5) The Nanobody binding in liver is assessed by SPECT imaging and postmortem ex vivo measurement in gamma counter. However, Im missing in vitro or ex vivo autoradiographic assessment of the binding in the liver. Is the liver uptake diffuse and homogenous (i.e. excretion) or localized to the diseased sinusoids (i.e. binding to the the inflamed regions). This is crucial information for assessing the value of the SPECT tracer.

6) Also, theres no assessment of the specificity of the binding. This should best case be performed by blocking VCAM-1 in vitro or in vivo, which should lead to a decreased binding of the 99Tc-Nanobody. An alternative is to use a control non-binding size matched Nanobody as comparator. A control Nanobody is actually mentioned in the Methods, but not used in the actual study. Without an assessment of the specificity, any increase in uptake can be non-specific and due to changes in biodistribution, excretion etc.

7) Related to above, the entire biodistribution in more tissues than just liver should be shown in all models. For example, liver uptake may be increased measured as SUV, but this could just follow from an increase in SUV in blood. See for example in Supplementary Figure 1 – SUV in blood is actually increased more than in liver in the V-MCD diet group! I would suspect the increased liver SUV is non-specific and not mediated by VCAM-1 binding. Similarly, spleen uptake could be a good positive control tissue with known expression of VCAM-1 in normal physiology.

DISCUSSION

1) The limitations outlined above must be discussed in detail. It has not been established that the liver uptake of this 99Tc-Nanobody is even mediated by binding to the VCAM-1 due to the lack of blocking, autoradiography matching with IHC etc etc.

Minor comments

Page 4. Please explain the different mouse model in some more detail, especially the FOZ model (characteristics?)

Perhaps make a table summarizing the different models used in the different studies (7 different models) – abbreviations, how many mice each etc. This would make the paper much easier to read and interpret.

Reviewer #3 (Remarks to the Author):

Comments for Authors

The authors describe a preclinical evaluation of an anti-VCAM-1 nanobody 99mTc-cAbVCAM1-5 for imaging of liver inflammation. The paper provides a rationale for development of diagnostic methodology, which potentially permits identify patients with a high degree of liver inflammatory activity ahead of the development of severe fibrosis. Thus, it address an unmet clinical need. The molecular biology basis for such approach is well presented. A serious advantage of the proposed tracer (in comparison with tracers developed for similar purpose) is its low uptake in normal liver. The study is thoroughly planned and carefully executed. The study methodology is sound. The

models are well-validated. All necessary controls are included. The figures are clear and present both average and individual values, as well as p-values when statistical treatment was performed. The methodology is described in sufficient details to be reproduced.

Overall, the study presents a clear preclinical rationale for further clinical development of a novel method for detection and progression monitoring of non-alcoholic fatty liver disease. In the case of success, this might be a substantial improvement of diagnostics and therapy response monitoring of quite widespread disease.

The authors should address some minor editorial comments.

Rows 82-85 "

" At the molecular level, this recruitment is driven by

83 attracting molecules and chemokines such as monocyte chemoattractant protein MCP-1.11 In
84 a coordinated manner, the adhesion molecules amongst which ICAM-1, vascular adhesion
85 protein-1 (VAP-1) and VCAM-1 promote their adhesion to the endothelium and their
86 migration to the liver."

Comment; Please rephrase. As it is currently written, the adhesion molecules (ICAM-1, vascular adhesion protein-1 (VAP-1) and VCAM-1) promote adhesion to the endothelium and migration to the liver of attracting molecules and chemokines.

Row 114/Figure 1 Please define the abbreviation STD at the first mentioning in the text.

Row 122/Figure 1 Please define the abbreviation V-MCD at the first mentioning in the text.

Figure 1 Please define the abbreviation C-MCD

Row 126 Please define the abbreviation ICC at the first mentioning in the text.

Row 161 Please define the abbreviation CLS at the first mentioning in the text.

Point-by-point response: "Molecular imaging of liver inflammation using an anti-VCAM-1 nanobody" (NCOMMS-22-08977).

We thank the Reviewers for the high-quality feedback that they provided. Their suggestions were excellent and insightful. They led to a thorough re-shuffling of our manuscript, relevant additional experiments and analysis that we believe have strengthened our key message.

Point-by-point response to Reviewer #1

In this study the authors determine if the molecular imaging von VCAM-1 could be used as an alternative non-invasive tool to detect liver inflammation in the setting of chronic liver disease. Key findings include that in various preclinical models of liver disease the uptake correlated with liver inflammation and molecular markers of inflammation. While the authors present very interesting data there are several issues regarding the study design and interpretation of the data that need to be addressed.

Proof of concept study: It's not clear why MCP-1 was chosen. Additional marker like NAS and infiltrating neutrophils that are related to NAFLD and NASH should be presented in addition to bolster the conclusion that VCAM-1 might be a marker to detect inflammation. Also, how about AST? And VCAM protein or mRNA?

Response: We thank the reviewer for his comment. In the proof-of-concept part, our aim was to evaluate the ability of the ^{99m}Tc -cAbVCAM1-5 to detect liver parenchymal inflammation. The preclinical model employed was MCD diet-fed mice as mean to induce liver steatosis and significant parenchymal inflammation, rapidly reversible upon switch to a normal diet feeding. However, MCD diet fed animals are not obese nor dysmetabolic (thus obesity and altered metabolism are not the drivers of liver pathology).¹ Also, they do not present hepatocyte ballooning, a stigmata of cell death seen in human NASH. Thus, *stricto sensu* they develop a form of steatohepatitis but not NASH. Hence, evaluating NAS in this particular model would be of low relevance. So, we report on steatosis and inflammation in the MCD model. By contrast, we reported detailed histological data (Fig. 3C-E) and comprehensive histological illustration for more relevant models of NAFLD/NASH (New Fig. 3G) that were employed in the main study. In addition, we now report the NAS in the main figure (New Fig. 3F).

Figure 3C-E

New Figure 3F

Regarding circulating enzymes specific to liver damage (i.e. alanine aminotransferases; ALT) and VCAM-1 mRNA levels in the MCD models, both data are reported in the Fig. 1C and Fig. 1E (see below) and were, in line with the literature², significantly higher in MCD-fed mice when compared to chow-fed controls.

New Figure 3G

Legend. Representative liver histology in all groups (thick arrow = inflammatory loci, thin arrow = ballooning)

Figure 1C

Figure 1E

How is the detection affected by the presence of fibrotic alterations? Please show some data from models with later stages of the disease, preferable the same model (not MCD model) at different stages of the disease.

Response: We thank the reviewer for raising this important point. As we previously reported³, at the time points analyzed (12 weeks HFD feeding in *foz/foz* mice and 4 weeks CDAA-HF feeding in C57BL6/j), pericellular fibrosis is present but is rather mild (Suppl. Fig. 4B). Furthermore, we find no or weak correlations between fibrosis degree and VCAM-1 protein content, VCAM-1 SUV Liver, VCAM-1 SUV_{Lean} Liver (see answer to Reviewer 2) and VCAM-1 %ID Liver (Suppl. Fig. 6A-D)

Supplementary Figure 4B

Supplementary Figure 6A

New Supplementary Figure 6B

New Supplementary Figure 6C

We performed a novel multivariate linear regression on the entire mice cohort and show that there is no association between the signal for VCAM uptake (whether evaluated with SUV, SUV_{Lean} or %ID) and fibrosis, irrespective of the model (New Suppl. Table 5).

New Supplementary Table 5

Characteristic	SUV			SUV _{Lean}			%ID		
	Beta	95% CI [†]	p-value	Beta	95% CI [†]	p-value	Beta	95% CI [†]	p-value
Ballooning									
0	—	—		—	—		—	—	
1	0.13	-0.06, 0.33	0.2	0.14	-0.14, 0.41	0.3	0.39	-0.45, 1.2	0.3
2	-0.25	-0.36, -0.15	<0.001	-0.35	-0.50, -0.21	<0.001	1.5	1.0, 1.9	<0.001
Inflammation									
0	—	—		—	—		—	—	
1	0.07	-0.02, 0.17	0.11	0.07	-0.06, 0.20	0.3	-0.05	-0.44, 0.35	0.8
2	0.26	0.12, 0.40	<0.001	0.37	0.17, 0.56	<0.001	-0.03	-0.62, 0.57	>0.9
3	0.41	0.32, 0.51	<0.001	0.63	0.50, 0.77	<0.001	0.81	0.40, 1.2	<0.001
Fibrosis	0.02	-0.03, 0.07	0.5	0.01	-0.06, 0.08	0.8	-0.15	-0.36, 0.06	0.2

[†] CI = Confidence Interval
Fibrosis is expressed in % of stained area

Legend. Multivariate linear regression to identify histological predictor of VCAM-1 signal (quantified with SUV, SUV_{Lean} or %ID of the liver in vivo). Inflammation and ballooning, but not fibrosis, are histological predictors of VCAM-1 signal (n=32).

In the same vein, human data support that VCAM-1 upregulation is specific to liver inflammation and maintained with fibrosis progression.⁴ Indeed, in the JCI paper by *Furuta et. al*⁴ to which we refer to in our manuscript, the expression of VCAM-1 according to NAFLD spectrum was investigated in mice and in patients. First, and in line with our observations, VCAM-1 was strongly upregulated in CDH mice, as well as in an (alternative) preclinical model of NASH, i.e. C57BL6/J fed a fat, fructose and cholesterol-rich diet. In patients with NAFLD, VCAM-1 was specifically upregulated in those with NASH - independently from the fibrosis stage (Figure 3D in *Furuta et. al*⁴). By contrast, VCAM-1 was not upregulated in patients with NAFLD but isolated steatosis (IS) (Figure 3D in *Furuta et. al*⁴). The human cohort was well balanced (Supplementary Table 4 in *Furuta et. al*⁴) and authors did not report any sex-difference in VCAM-1 expression according to NAFLD status.

Figure 3D (from *Furuta et. al⁶*)

Legend. Quantification of VCAM-1 staining of liver tissue sections obtained from patients with normal liver (NL), isolated steatosis (IS), and NASH with fibrosis stages 0–1 (F0–1) or 2–4 (F2–4).

Supplementary Table 4 (from *Furuta et. al⁶*)

Patients' characteristics in each category based on liver histology.

	Age (years)	Gender (M/W)	BMI (kg/m ²)	AST (U/L)	ALT (U/L)	Mode of tissue acquisition
Normal Liver	57	M	31.8	15	17	HR
	56	M	34.9	38	21	HR
	69	M	34.2	8	N.D.	HR
	38	F	34.9	28	24	HR
	55	F	45.4	51	24	HR
	78	F	30.7	18	23	HR
Isolated Steatosis	47	M	40.6	72	23	HR
	46	M	45.6	39	19	HR
	29	F	46.5	N.D.	13	HR
	56	F	50.7	23	98	HR
	48	F	42.9	16	15	HR
	63	F	39.5	29	22	HR
	40	M	44.8	N.D.	47	HR
NASH F0-1	49	F	52.1	N.D.	23	HR
	60	F	37.8	11	28	HR
	36	M	39.1	68	39	HR
	45	M	31.3	30	24	HR
	52	M	38.2	127	56	HR
	27	F	40.6	13	13	HR
	26	M	63.2	63	40	HR
NASH F2-4	44	M	45.0	67	55	LB
	58	M	21.0	22	86	HR
	63	F	62.0	44	21	HR
	67	F	45.8	78	83	LB
	56	F	40.3	20	21	HR

HR: hepatic resection; LB: liver biopsy

Taken collectively, these evidences support that VCAM-1 expression is specific to (steato)hepatitis, i.e. parenchymal inflammation in a fatty liver (a key characteristic which, as highlighted by Reviewer 3, distinguishes it from other imaging probes tried so far) and that its high expression in the liver is maintained with fibrosis progression.

Nonetheless, as the Reviewer pointed out, we must acknowledge the possibility that in end-stage fibrosis / cirrhosis, the VCAM-1 signal could be distorted, as a consequence of reduced accession of the nanobody to the sinusoidal endothelium and transformation (capillarization) of the endothelium. This question was not addressed in our study because there are efficient (point of care) noninvasive tools to identify end-stage liver disease, e.g. elastography or blood-based scores.⁵ This point is now thoroughly discussed in the manuscript (p. 11-12).

Thus, we anticipate that VCAM-1 imaging – even if proven accurate in advanced fibrosis/cirrhosis - will be of moderate added value in this particular context. By striking contrast, there is still no way to detect which of the billions of individuals with NAFLD have a high degree of liver inflammatory in which fibrosis is not yet advanced (i.e. NASH F0-1). The latter represent half of the total NASH population⁶ - hence hundreds of millions of individuals worldwide. These patients might progress to cirrhosis and eventually develop HCC^{7,8}, but will typically remain unaware of their condition until critical disease stages. Given that a liver biopsy cannot be realistically proposed to screen such a large population, our point was to develop a non-invasive method to assess the disease activity and give the opportunity for early intervention(s) - well-ahead of advanced fibrosis/cirrhosis. Therefore, we are now focusing our effort on efficiently translating these findings to the clinic, aiming at detecting NASH in at-risk populations (e.g. subjects with type 2 diabetes). This point is now better discussed in the manuscript (p. 11)

The representative pictures shown of the histology of livers are rather poor in quality. And while this is not the main focus of the study, better pictures would be very beneficial. Also please provide more characterization of the models used in regard to liver damage and inflammation.

Response: We agree with the Reviewer and apologize for the low quality of histology in the proof-of-concept part. This is explained by the availability of only flash-frozen liver for cryosection in the MCD experiment. As elaborated above, we focused on molecular marker of inflammations and specificity of the nanobody in this proof-of-concept part, while leaving detailed liver histology evaluation for the more relevant NAFLD/NASH models in the main part of our manuscript. In the latter, the histological and molecular phenotyping of liver disease were exhaustive, i.e. detailed NAS (Fig. 3C-F), new histological illustrations (*see above* New Fig. 3G), key inflammatory genes (Fig. 4A-B), steatosis and crown-like structures quantification (Suppl. Fig. 2) and fibrosis quantification (Suppl. Fig. 4). In order to meet the high-quality standards of *Nature Communications*, we propose to include novel histological illustrations for all NAFLD models of the main study (new Fig. 3G)

How is VCAM-1 expression in different stages of NAFLD in humans?

Response: We thank the reviewer for this important translational question and refers to our reply to the precedent comments, wherein we believe it was addressed thoroughly.

Are there any adverse effects of the labeled VCAM-1?

^{99m}Tc-cAbVCAM1-5 has been previously employed in mice models of atherosclerosis⁹ and a phase I/IIa clinical trial (NCT04483167), mainly aimed at evaluated its safety and dosimetry, is currently being conducted at Grenoble-Alpes University Hospital (NCT04483167). In order to initiate this clinical trial, regulatory single dose extended toxicity study has been performed by a certified CRO (CERB: Centre de Recherches Biologiques) and included in the Investigational Medicinal Product Dossiers (IMPD). The conclusion was that, under the experimental conditions adopted, test item ^{99m}Tc-cAbVCAM1-5 administered by the intravenous route as a bolus induced no mortality, no sign of toxicity and was well tolerated in the male or female

mice. In term of dosimetry, ^{99m}Tc is routinely employed in clinical practice for SPECT nuclear imaging, and since ^{99m}Tc -cAbVCAM1-5 main accumulation is observed in the renal cortex, it was anticipated that its dosimetry will be in line with that of ^{99m}Tc -DMSA that is routinely employed in clinical practice to evaluate kidney function.

In accordance with these previous results, no signs of toxicity, such as weight loss, was observed in the present study.

Line 138: soften the statement: liver inflammation in an MCD-fed mouse. We reformulated the sentence.

Point-by-point response to Reviewer #2

General comments:

New endpoints to assess and quantify liver inflammation is an urgent clinical need, given the increase in NAFL and diabetes worldwide. Thus, accurate diagnostics are welcome. Furthermore, novel interventions require sensitive, repeatable and non-invasive assessment of treatment effect – this is certainly lacking today which is a major obstacle for drug development and approval.

Nachit et al has repurposed a novel Nanobody binder to VCAM-1, with the hypothesis that non-invasive SPECT imaging of this protein would provide a suitable biomarker for liver inflammation. Nanobodies are small (≈ 15 kDa) proteins which theoretically are excellent scaffolds for imaging agents.

The study is well motivated and interesting but with major limitation and quite a lot of missing background information which makes it difficult to assess and interpret.

Response: We thank the reviewer for his comments and fully agree with the need for accurate diagnosis in NAFLD.

Major comments:

1) The nanobody ^{99m}Tc -cAbVCAM1-5 is poorly described which makes it very difficult to interpret the results. What is the affinity to human VCAM-1? Murine VCAM-1. Seems to be \approx nM when searching the literature, but this should be explicitly stated. What is the stability in vivo and in plasma?

Response: These data are indeed available in our previous publications.^{9,10} We now give a summary in the introduction for clarity purpose. The introduction is amended as follows (changes in red). We also comment on toxicity and specificity:

“[...] we employed an anti-VCAM-1 nanobody labelled with technetium-99m (^{99m}Tc -cAbVCAM1-5), that is currently undergoing clinical evaluation for atherosclerosis imaging, in various preclinical models of NAFLD. This imaging agent is a cross reactive binder of murine and human VCAM-1 proteins with nanomolar affinity as determined by surface plasmon resonance ($KD = 2.0 \pm 0.0$ nM for murine and 6.5 ± 0.7 nM for human VCAM-1). ^{99m}Tc -cAbVCAM1-5 was found to be stable in vivo in mouse blood up to 3 hours. It specifically binds to VCAM-1 as demonstrated in vivo using an excess of cold competitor. Furthermore, ^{99m}Tc -cAbVCAM1-5 was found to be well tolerated and exclusively eliminated through the kidneys, suggesting that it could be employed for inflammation imaging in the liver⁹

2) The pilot and main studies are analysed using different endpoints (SUV and %ID respectively). In the end %ID is recommended by the authors. The %ID should be calculated also for the pilot study, and correlated to other assessment of inflammation, like in the main study.

Response: Please see joint answer below.

3) It's unclear why %ID would be a better endpoint than SUV (or %ID/g). SUV or %ID/g normally represents the concentration of the SPECT tracer or the target receptor – and I assume that we would like to measure the density of the liver inflammation, i.e. the “amount of inflammatory activity” per liver volume? %ID would measure the total amount of inflammation in a whole liver, but this would lead to potentially very strange results. Consider two persons, who both have a liver uptake of SUV = 1. Person 1 is tall with a large liver of 2.5 litres. Person 2 is a smaller person with a liver volume of 1.25 litres. Neither has NAFL. But %ID will be 2 times

higher in person 1 – even though only the liver size is larger, not the concentration of inflammation.

I understand the attempt to correct for increased steatosis, but this must be done in a more advanced manner, for example by measuring steatosis by MR or similar.

Response: We thank the reviewer for these truly insightful comments and propose to answer to comments 2) and 3) jointly. Indeed, by using %ID/organ, our attempt was to correct the uptake of the radiotracer for steatosis severity: steatosis is dead volume and commonly used units (%ID/g and SUV) do not correct for the dead volume induced by steatosis, while %ID/organ is not influenced by this parameter and therefore better reflect the global inflammatory state. We do not fully agree with the reviewer regarding the above-mentioned example as SUV is corrected both for ID and for total body weight, and therefore for the changes in distribution volumes. Hence assuming two healthy persons (tall and small) with a liver representing ~2% of their TBW, then both their SUV and %ID/organ should be similar. For this reason, we posited that the %ID/organ could correct for steatosis-induced dilution of lean liver uptake to some extent, but we fully concur that it cannot control for both steatosis and other parameters that could affect the liver to TBW ratio (which can indeed be modified in NAFLD as illustrated in the preclinical models).

Hence, we followed the excellent suggestion of the reviewer and attempted to correct SUV for liver steatosis. This was entirely feasible as we had the precise histological quantification of steatosis percentage and the liver density in Hounsfield Unit (HU) (inversely correlated to steatosis degree¹¹) to correct for SUV ex-vivo and in-vivo, respectively.

We first calculated the lean liver proportion as 100 % – steatosis % based on histological data, e.g. 80% if steatosis was quantified at 20%. When then divided the SUV_{ex-vivo} by the lean liver proportion to adjust for the steatosis-induced dead volume, irrespective of the total liver size or the %ID. The so corrected SUV (SUV_{Lean Liver}_{ex-vivo}) was the highest in animals with steatohepatitis and was highly correlated with the histological inflammatory score (r=0.89, p<0.0001), validating the approach.

New Fig. 5E

Legend. SUV_{Lean Liver}_{ex-vivo} in all groups (one-way ANOVA and Student's t-test, n=6-8 mice/group)

Legend. Spearman's correlation between SUV_{Lean Liver}_{ex-vivo} and histological inflammatory stages

Next, we aimed to adapt this methodology in vivo. To do so, we first verified that liver density in HU was correlated with histological steatosis. We found a high correlation, especially for steatosis > 5% (i.e. the definition of NAFLD). Of note, CT HU values exhibit a large variability when steatosis <5%. This relationship was expected from the literature, and the curve reminiscent of validation studies in humans (see below). In a clinical setting, such cases would

however be rare in our target population – given that they would not qualify for a NAFLD diagnosis in the first place.

Figure 3A (from *Pickhardt et. al*¹²)

Legend. Plot of the primary cohort of patients (n=72)¹², where CT scanning was performed within one month of CSE-MRI, showing good linear correlation (r²=0.828). This cohort was utilized to derive the linear conversion formula: $MRI-PDFF(\%) = -0.58 \times CT(HU) + 38.2$. Correlation significantly drops off at non-steatotic lipid levels (<5%).

New Suppl. Fig. 8

Legend. Pearson's correlation between SPECT-based liver density (HU) and histological liver steatosis. As expected from the literature¹², the correlation is very good for steatosis > 5% and drops at non-steatotic lipid levels.

We found that all mice with $\geq 5\%$ of liver steatosis had a liver density < -50 HU. Thus, we corrected for steatosis when liver density is < -50 HU in the present study, as correcting for 1-5% steatosis would have a minimal impact on SUV (and would bias the linear regression equation). The following formula was derived: $\text{liver steatosis} (\%) = -0.2476 \times \text{Liver HU} - 3.945$. We corrected the $SUV_{\text{Lean Liver}_{\text{in-vivo}}}$ with CT-based liver steatosis (%), yielding a $SUV_{\text{Lean Liver}_{\text{in-vivo}}}$. $SUV_{\text{Lean Liver}_{\text{in-vivo}}}$ was strongly correlated with the histological inflammatory score (r = 0.89, p<0.0001) and with VCAM-1 protein concentration in the liver.

New Fig. 5D

Legend. $SUV_{\text{Lean Liver}_{\text{in-vivo}}}$ in all groups (one-way ANOVA and Student's t-test, n=6-8 mice/group)

New Suppl. Fig. 9A

Legend. Spearman's correlation between $SUV_{\text{Lean Liver}_{\text{in-vivo}}}$ and histological inflammatory score

Finally, we investigated the diagnostic power of $SUV_{\text{Lean Liver}_{\text{in-vivo}}}$ to detect any level of inflammation (1 to 3). The AUCs in the whole cohort or only in mice with NAFLD were excellent (0.94 or 0.86, respectively, p<0.05). We evaluated the diagnostic power to detect moderate-

to-high inflammation, as it is probably more translationally relevant (i.e. the clinicians need to detect the more active cases - at the highest risk of progression¹³⁻¹⁶). The test was almost perfectly discriminant in the whole cohort and excellent when we only consider mice with NAFLD (AUCs of 0.98 and 0.95, respectively, $p < 0.05$). Of note, we did not elaborate on quantification methods in the proof-of-concept part since the MCD model has limited translational relevance as elaborated above (e.g. liver weight decreases which is unusual before cirrhotic stages).

In a clinical context, we might encounter different scenarios such as NASH with various Liver/TBW ratio and steatosis severity. Based on our preclinical data, we cannot ascertain the superiority of one corrective methods. It is however true that SUV_{Lean} will be convenient for clinical practice, especially given than formulae to evaluate liver steatosis based on HU have been developed and validated in large cohorts.^{12,17-19} Alternatively, the technique could also be implement with MRI-based nuclear imaging techniques as sequences yielding a proton density fat fraction (PDFF) map are validated, fast and widely available.

We again thank the reviewer for this brilliant suggestion and propose to include both methods in main figures of the manuscript (Fig. 5 and Fig. 6), further discussing that clinical studies will help determine the most accurate in a real-life scenario.

New Fig. 9A

Legend. ROC curves for detection of liver inflammation (0-1 vs 2-3, red and 0 vs 1-3, blue) based on SUV_{Lean} $Liver_{in-vivo}$ in the entire cohort, i.e. NAFLD + normal liver (n=32).

New Fig. 9C

Legend. ROC curves for detection of liver inflammation (0-1 vs 2-3, red and 0 vs 1-3, blue) based on SUV_{Lean} $Liver_{in-vivo}$ in mice with NAFLD (n=22).

4) Is there a reason why VCAM-1 wasn't stained for on sections? Other inflammation/macrophage markers were used instead. The authors should comment and explain this decision.

Response: Our goal, when evaluating VCAM-1 expression in the liver, was to obtain quantitative data that could then be compared to the uptake of the evaluated imaging agent. For this reason, we evaluated VCAM-1 mRNA with PCR and protein levels with ELISA, but did not perform immunohistological assessment of VCAM-1. IHC is the method of choice for identifying the location of a target protein. However, VCAM-1 being known to be diffusely expressed within the entire liver parenchyma in murine models of NAFLD (<https://www.jci.org/articles/view/143690/figure/3>), we did not anticipate focal expression that could then be compared to ^{99m}Tc -cAbCAM1-5 distribution, either determined in vivo by SPECT or ex vivo by autoradiography, since none of this technique offers a sufficient spatial resolution to identify microvessels (*see also question 5 below*).

5) The Nanobody binding in liver is assessed by SPECT imaging and postmortem ex vivo measurement in gamma counter. However, I'm missing in vitro or ex vivo autoradiographic assessment of the binding in the liver. Is the liver uptake diffuse and homogenous (i.e. excretion) or localized to the diseased sinusoids (i.e. binding to the the inflamed regions). This is crucial information for assessing the value of the SPECT tracer.

Response: Autoradiography was performed on a subset of livers. ^{99m}Tc -cAbCAM1-5 distribution was found to be homogenous. This was anticipated because VCAM-1 is expressed by the endothelium of the liver microvasculature, a target too small to be identified using autoradiography that has a resolution in the 100 μm range. All of the animal models that have been employed in the present study have previously been well characterized using histology and immunohistology techniques^{11,20}, and inflammation is known to occur in all parts of the liver. This was further confirmed here by F4/80 and HE staining examination. We therefore concluded that, due to the small size of the target and of its homogenous distribution throughout all the liver, autoradiography combined with VCAM-1 staining on an adjacent tissue slice was not a valid approach to confirm the binding of ^{99m}Tc -cAbVCAM1-5 to its target. We however agree with the reviewer on the fact that this is a crucial parameter to investigate when developing an imaging agent. Therefore, in order to demonstrate that this homogeneous liver uptake of ^{99m}Tc -cAbVCAM1-5 was attributable to a diffuse expression of VCAM-1 throughout the liver, we performed a new in vivo competition study (*see question 6 below*). Regarding the visualization of the cellular localization of cAbVCAM1-5, that is a project we are currently developing in our laboratory. We have recently initiated the production of a fluorescent version of cAbVCAM1-5. The ultimate goal will be to inject the probe in vivo in various animal models and then to investigate its distribution on tissue slices using confocal microscopy. Such an approach will allow to circumvent the limitation of autoradiography.

6) Also, there's no assessment of the specificity of the binding. This should best case be performed by blocking VCAM-1 in vitro or in vivo, which should lead to a decreased binding of the ^{99m}Tc -Nanobody. An alternative is to use a control non-binding size matched Nanobody as comparator. A control Nanobody is actually mentioned in the Methods, but not used in the actual study. Without an assessment of the specificity, any increase in uptake can be non-specific and due to changes in biodistribution, excretion etc.

Response: We agree with the Reviewer that a competitive assay is important to evaluate the specificity of ^{99m}Tc -cAbVCAM1-5. In our study, we used a non-specific nanobody (cAbBcl110) radiolabeled with ^{99m}Tc in the MCD model to control for irrelevant binding to the liver. To

complement this approach, we now perform and report the data from an additional competition experiment in CDH-fed mice. We chose the CDH model as steatohepatitis is conveniently induced in 4-weeks, allowing us to meet the time requirements for the present re-submission.

Hence, ^{99m}Tc -cAbVCAM1-5 (5 μg) was either injected alone or in combination with a 100x excess of cold cAbVCAM1-5 (500 μg) (n=5 per group). As supported by the data (see below, new Suppl. Fig. 5), the binding of cAbVCAM1-5 in a liver with steatohepatitis was found to be highly specific. Indeed, the signal was blunted when ^{99m}Tc -cAbVCAM1-5 was co-injected with the 100x excess cold nanobody. Image quantification confirmed that SUV was of 0.89 ± 0.10 in the group without competitor and that the competition resulted in a significant decrease to 0.21 ± 0.03 (mean \pm SD; P=0.0079 using Mann and Whitney), a value similar to that previously observed in control animals or when using an irrelevant nanobody.

New Suppl. Fig. 5

Legend. *In vivo competition study.* ^{99m}Tc -cAbVCAM1-5 was injected in CDH-fed mice either alone (n=5) or together with a 100-fold excess of unlabeled competitor cAbVCAM1-5 (n=5). Top: Quantification of liver SPECT imaging; Bottom: Representative liver SPECT/CT images. Competition resulted in a significant decrease of ^{99m}Tc -cAbVCAM1-5 uptake in liver (~76.4%), thereby demonstrating specificity of the signal. P=0.0078 versus no competition

7) Related to above, the entire biodistribution in more tissues than just liver should be shown in all models. For example, liver uptake may be increased measured as SUV, but this could just follow from an increase in SUV in blood. See for example in Supplementary Figure 1 – SUV in blood is actually increased more than in liver in the V-MCD diet group! I would suspect the increased liver SUV is non-specific and not mediated by VCAM-1 binding. Similarly, spleen uptake could be a good positive control tissue with known expression of VCAM-1 in normal physiology.

Response: We agree with the Reviewer and now provide detailed biodistribution for all models in Supplemental Tables 1, 3 and 4. It is true that in both MCD and FH models, blood activity - when expressed as an SUV - is found to be higher than in the respective control groups. In both models however, blood SUV is lower to that of liver, so that the increase observed in the liver cannot solely originate from the blood volume within the liver. In the CDH model of NAFLD, blood activity is similar to that observed in the control CSH group (p=NS) while a significant increase is observed in the liver (p<0.0001), thereby ruling out the hypothesis the observed increase could be attributable to a non-specific accumulation. Taken together with the in vivo competition experiment provided above, we hope that we convince the reviewer that VCAM-1 binding in the liver is specific. We have considered using the spleen as an internal control in previous studies. However, it appears that VCAM-1 expression in lymphoid organs, while being

constitutively observed in healthy animal, can also be significantly increased in various pathophysiological conditions. This is likely in line with a variable degree of ‘systemic’ inflammation. Importantly, low grade systemic inflammation is a pathophysiological cornerstone in conditions associated with obesity and metabolic syndrome.²¹

DISCUSSION

1) The limitations outlined above must be discussed in detail. It has not been established that the liver uptake of this 99Tc-Nanobody is even mediated by binding to the VCAM-1 due to the lack of blocking, autoradiography matching with IHC etc etc.

We thank the reviewer for his very constructive comments and hope to have address his concerns in the present version of the manuscript. We thoroughly re-worked the results section, integrating the new data derived from steatosis-corrected SUV (SUV_{Lean} Liver) (p. 8-9), explaining the rationale in the methodology section (p. 14) and discussing them in a neutral manner (p. 11).

Minor comments

Page 4. Please explain the different mouse model in some more detail, especially the FOZ model (characteristics?) *Done (see below)*

Perhaps make a table summarizing the different models used in the different studies (7 different models) – abbreviations, how many mice each etc. This would make the paper much easier to read and interpret. *Done (New Supplementary Table 2)*

Manuscript part	Genetic background	Genotype	Diet	Abbreviation	Number of animals	NAFLD	Steatohepatitis	Metabolic syndrome
Proof-of-concept	C57Bl6/j	Wild-type	Methionine-choline deficient	MCD	20	-	Severe	Absent
	C57Bl6/j	Wild-type	Standard chow	STD	10	-	Absent	Absent
Main study	NOD.B10	Wild-type	Standard chow	Ctl	8	Absent	Absent	Absent
	NOD.B10	Wild-type	High fat	WH	6	Mild	Mild	Mild
	NOD.B10	FOZ	High fat	FH	6	Severe	Severe	Severe
	C57Bl6/j	Wild-type	Choline-supplemented high fat	CSH	6	Mild	Mild	Mild
	C57Bl6/j	Wild-type	Choline-deficient high fat diet	CDH	6	Severe	Severe	Absent

Point-by-point response to Reviewer #3

Comments for Authors

The authors describe a preclinical evaluation of an anti-VCAM-1 nanobody 99mTc-cAbVCAM1-5 for imaging of liver inflammation. The paper provides a rationale for development of diagnostic methodology, which potentially permits identify patients with a high degree of liver inflammatory activity ahead of the development of severe fibrosis. Thus, it address an unmet clinical need. The molecular biology basis for such approach is well presented. A serious advantage of the proposed tracer (in comparison with tracers developed for similar purpose) is its low uptake in normal liver.

The study is thoroughly planned and carefully executed. The study methodology is sound. The models are well-validated. All necessary controls are included. The figures are clear and present both average and individual values, as well as p-values when statistical treatment was performed. The methodology is described in sufficient details to be reproduced.

Overall, the study presents a clear preclinical rationale for further clinical development of a novel method for detection and progression monitoring of non-alcoholic fatty liver disease. In the case of success, this might be a substantial improvement of diagnostics and therapy response monitoring of quite widespread disease.

We thank the Reviewer for his very positive evaluation of our manuscript and have addressed all minor comments in the present version.

The authors should address some minor editorial comments:

Rows 82-85 “

“ At the molecular level, this recruitment is driven by 83 attracting molecules and chemokines such as monocyte chemoattractant protein MCP-1.11 In 84 a coordinated manner, the adhesion molecules amongst which ICAM-1, vascular adhesion 85 protein-1 (VAP-1) and VCAM-1 promote their adhesion to the endothelium and their86 migration to the liver.” .

Comment; Please rephrase. As it is currently written, the adhesion molecules (ICAM-1, vascular adhesion protein-1 (VAP-1) and VCAM-1) promote adhesion to the endothelium and migration to the liver of attracting molecules and chemokines. Done

Row 114/Figure 1 Please define the abbreviation STD at the first mentioning in the text. Done

Row 122/Figure 1 Please define the abbreviation V-MCD at the first mentioning in the text. Done

Figure 1 Please define the abbreviation C-MCD Done

Row 126 Please define the abbreviation ICC at the first mentioning in the text. Done

Row 161 Please define the abbreviation CLS at the first mentioning in the text. Done

Vladimir Tolmachev

References

1. Farrell G, Schattenberg JM, Leclercq I, et al. Mouse Models of Nonalcoholic Steatohepatitis: Toward Optimization of Their Relevance to Human Nonalcoholic Steatohepatitis. *Hepatology*. 2019;69(5):2241-2257. doi:10.1002/hep.30333
2. Machado MV, Michelotti GA, Xie G, et al. Mouse models of diet-induced nonalcoholic steatohepatitis reproduce the heterogeneity of the human disease. *PLoS ONE*. 2015;10(5):1-16. doi:10.1371/journal.pone.0127991
3. De Rudder M, Bouzin C, Nachit M, et al. Automated computerized image analysis for the user-independent evaluation of disease severity in preclinical models of NAFLD/NASH. *Lab Invest*. 2020;100(1):147-160. doi:10.1038/s41374-019-0315-9
4. Furuta K, Guo Q, Pavelko KD, et al. Lipid-induced endothelial vascular cell adhesion molecule 1 promotes nonalcoholic steatohepatitis pathogenesis. *J Clin Invest*. 2021;131(6). doi:10.1172/JCI143690
5. Castera L, Friedrich-Rust M, Loomba R. Noninvasive Assessment of Liver Disease in Patients With Nonalcoholic Fatty Liver Disease. *Gastroenterology*. 2019;156(5):1264-1281.e4. doi:10.1053/j.gastro.2018.12.036
6. Dufour JF, Scherer R, Balp MM, et al. The global epidemiology of nonalcoholic steatohepatitis (NASH) and associated risk factors—A targeted literature review. *Endocr Metab Sci*. 2021;3(February). doi:10.1016/j.endmts.2021.100089
7. Anstee QM, Reeves HL, Kotsiliti E, Govaere O, Heikenwalder M. From NASH to HCC: current concepts and future challenges. *Nat Rev Gastroenterol Hepatol*. 2019;16(7):411-428. doi:10.1038/s41575-019-0145-7
8. Huang DQ, El-Serag HB, Loomba R. Global epidemiology of NAFLD-related HCC: trends, predictions, risk factors and prevention. *Nat Rev Gastroenterol Hepatol*. 2021;18(4):223-238. doi:10.1038/s41575-020-00381-6
9. Broisat A, Hernot S, Toczek J, et al. Nanobodies Targeting Mouse/Human VCAM1 for the Nuclear Imaging of Atherosclerotic Lesions. *Circ Res*. 2012;110(7):927-937. doi:10.1161/CIRCRESAHA.112.265140
10. Broisat A, Toczek J, Dumas LS, et al. 99mTc-cAbVCAM1-5 Imaging Is a Sensitive and Reproducible Tool for the Detection of Inflamed Atherosclerotic Lesions in Mice. *J Nucl Med*. 2014;55(10):1678-1684. doi:10.2967/jnumed.114.143792
11. De Rudder M, Bouzin C, Nachit M, et al. Automated computerized image analysis for the user-independent evaluation of disease severity in preclinical models of NAFLD/NASH. *Lab Invest*. 2020;100(1):147-160. doi:10.1038/s41374-019-0315-9
12. Pickhardt PJ, Graffy PM, Reeder SB, Hernando D, Li K. Quantification of Liver Fat Content With Unenhanced MDCT: Phantom and Clinical Correlation With MRI Proton Density Fat Fraction. *Am J Roentgenol*. 2018;211(3):W151-W157. doi:10.2214/AJR.17.19391
13. Argo CK, Northup PG, Al-Osaimi AMS, Caldwell SH. Systematic review of risk factors for fibrosis progression in non-alcoholic steatohepatitis. *J Hepatol*. 2009;51(2):371-379. doi:10.1016/j.jhep.2009.03.019
14. Pais R, Charlotte F, Fedchuk L, et al. A systematic review of follow-up biopsies reveals disease progression in patients with non-alcoholic fatty liver. *J Hepatol*. 2013;59(3):550-556. doi:10.1016/j.jhep.2013.04.027
15. McPherson S, Hardy T, Henderson E, Burt AD, Day CP, Anstee QM. Evidence of NAFLD progression from steatosis to fibrosing-steatohepatitis using paired biopsies: Implications for prognosis and clinical management. *J Hepatol*. 2015;62(5):1148-1155. doi:10.1016/j.jhep.2014.11.034
16. Singh S, Allen AM, Wang Z, Prokop LJ, Murad MH, Loomba R. Fibrosis Progression in Nonalcoholic Fatty Liver vs Nonalcoholic Steatohepatitis: A Systematic Review and Meta-analysis of Paired-Biopsy Studies. *Clin Gastroenterol Hepatol*. 2015;13(4):643-654.e9. doi:10.1016/j.cgh.2014.04.014
17. Pickhardt PJ, Park SH, Hahn L, Lee SG, Bae KT, Yu ES. Specificity of unenhanced CT for non-invasive diagnosis of hepatic steatosis: Implications for the investigation of the natural history of incidental steatosis. *Eur Radiol*. 2012;22(5):1075-1082. doi:10.1007/s00330-011-2349-2

18. Graffy PM, Sandfort V, Summers RM, Pickhardt PJ. Automated Liver Fat Quantification at Nonenhanced Abdominal CT for Population-based Steatosis Assessment. *Radiology*. 2019;293(2):334-342. doi:10.1148/radiol.2019190512
19. Graffy PM, Pickhardt PJ. Quantification of hepatic and visceral fat by CT and MR imaging: relevance to the obesity epidemic, metabolic syndrome and NAFLD. *Br J Radiol*. 2016;89(1062):20151024. doi:10.1259/bjr.20151024
20. Nachit M, De Rudder M, Thissen JP, et al. Myosteatorsis rather than sarcopenia associates with non-alcoholic steatohepatitis in non-alcoholic fatty liver disease preclinical models. *J Cachexia Sarcopenia Muscle*. 2021;12(1):1-15. doi:10.1002/jcsm.12646
21. Lefere S, Van De Velde F, Devisscher L, et al. Serum vascular cell adhesion molecule-1 predicts significant liver fibrosis in non-alcoholic fatty liver disease. *Int J Obes*. 2017;41(8):1207-1213. doi:10.1038/ijo.2017.102

REVIEWERS' COMMENTS

Reviewer #1 (Remarks to the Author):

The authors addressed all my comments.

Reviewer #2 (Remarks to the Author):

Thanks for the revised manuscript - the additional data and analyses makes the study much more convincing. All of my major comments and questions have been answered.